## Registered report

psychology/cognition/neuroscience

expectation, social decision-making, music, EEG

**Author for correspondence:**
Elisa Carrus
e-mail: carruse@lsbu.ac.uk

# Does unfairness sound wrong? A cross-domain investigation of expectations in music and social decision-making

## Claudia Civai, Rachel Teodorini and Elisa Carrus

Division of Psychology, School of Applied Sciences, London South Bank University, London, UK

 CC, 0000-0002-6745-2074; RT, 0000-0002-9983-7008; EC, 0000-0001-9197-8757

This study was interested in investigating the existence of a shared psychological mechanism for the processing of expectations across domains. The literature on music and language shows that violations of expectations produce similar neural responses and violating the expectation in one domain may influence the processing of stimuli in the other domain. Like music and language, our social world is governed by a system of inherent rules or norms, such as fairness. The study therefore aimed to draw a parallel to the social domain and investigate whether a manipulation of melodic expectation can influence the processing of higher-level expectations of fairness. Specifically, we aimed to investigate whether the presence of an unexpected melody enhances or reduces participants' sensitivity to the violations of fairness and the behavioural reactions associated with these. We embedded a manipulation of melodic expectation within a social decision-making paradigm, whereby musically expected and unexpected stimuli will be simultaneously presented with fair and unfair divisions in a third-party altruistic punishment game. Behavioural and electroencephalographic responses were recorded. Results from the pre-planned analyses show that participants are less likely to punish when melodies are more unexpected and that violations of fairness norms elicit medial frontal negativity (MFN)-life effects. Because no significant interactions between melodic expectancy and fairness of the division were found, results fail to provide evidence of a shared mechanism for the processing of expectations. Exploratory analyses show two additional effects: (i) unfair divisions elicit an early attentional component (P2), probably associated with stimulus saliency, and (ii) mid-value divisions elicit a late MFN-like component, probably reflecting stimulus ambiguity. Future studies could build on these results to further investigate the

effect of the cross-domain influence of music on the processing of social stimuli on these early and late components.

# 1. Introduction

The role that expectation and prediction play across all areas of cognition has been widely investigated, from perception and action to decision-making [1–3]. One way of studying expectations is by investigating systems that are highly structured and therefore governed by a system of rules. An example of one such system is music, which involves the combination of units into higher-order structures (e.g. notes into melodies) that evolve over time according to certain rules (e.g. harmony) [4]. Knowledge of these rules is fundamental for our understanding of these systems and informs our expectations. For example, we implicitly learn the rules of music through repeated exposure to musical stimuli [5], and priming studies show that we have strong expectations about incoming musical stimuli [6]. There has been an increasing interest in the study of expectations using music [7–9], and how it may apply to other domains. For example, in a similar way to music, language is governed by a system of rules (grammar) that allows for the combination of smaller elements into structures that unfold over time (words into sentences). Interestingly, studies have shown that violating expectations in music can influence the way we process expectations in language. For example, musical context can influence the processing of language stimuli whereby participants are faster in processing phonemes when these were sung on a syntactically expected compared to a less syntactically expected music chord [10]. This priming effect of the musical context occurs quite automatically and regardless of musical expertise, and has been reported with more complex linguistic computations. For example, participants show a stronger semantic priming (faster responses for semantically related versus unrelated targets) when words are presented with expected compared to unexpected musical chords [11]. Related to this, violations of structure in music and language produce similar neural responses. More specifically, event-related potential (ERP) studies show early anterior negativities in the electroencephalographic signal (EEG) in both domains: N1 and early right anterior negativity components are elicited following violations of expectations in music, (early) left anterior negativity components following violations of language syntax [8,12,13]. Interestingly, the simultaneous presentation of structural violations in the two domains results in an interaction: the amplitude of the LAN is significantly reduced when presented simultaneously with unexpected music stimuli [14,15]. Similarly, the behavioural literature shows that language expectancy effects (faster processing for expected versus unexpected sentences) are reduced when presented with an unexpected musical sequence [16]. Therefore, the literature shows that when rules that inform our expectations of these structures are violated in both music and language, we observe similar responses. Additionally, when violations occur concurrently in the two domains, interactions are observed. These two patterns of results have been taken to support the hypothesis that music and language may share access to a common, but limited, pool of resources, probably specific to the processing of structural information from which expectations are derived [17,18].

The above investigation of structural expectations in music and language offers a useful framework to investigate the role of expectations more generally within the cognitive system. Besides these two domains, other systems are inherently governed by rules that inform our expectations. For example, social systems are characterized by social norms, defined by the Stanford Encyclopaedia of Philosophy as 'the informal rules that govern behaviour in groups and society, [… and] the unplanned result of individuals' interaction' [19]. Social norms are derived from exposure to the social environment and social behaviours around us, which informs our expectations and guides social decision-making [19]. When social norms, such as fairness or trust and reciprocity, are violated, people often react against these violations in an attempt to re-establish a social equilibrium [20,21]. In general terms, we can draw a parallel between expectations in music and the social domain in that they are both governed by rules that are derived from exposure to the environment and inform our expectations, ultimately guiding our behaviour. One notable difference, however, is that the expectations in music are based on a system of rules that governs the way units are meaningfully combined into structures as they unfold over time (e.g. notes into melodies) [4]. This is not the case for expectations derived from social norms. Having said so, both systems are similar in that knowledge about the rules and norms in the two domains allows for the formation of expectations about what is to follow given a certain preceding context. In music, the listener may expect a certain note following a given melodic context;

in the same way, an observer may expect a specific behaviour given a certain social scenario (e.g. reciprocate trust or re-establish fair outcomes). Given the above, the present study aims to investigate whether violating expectations in music will influence the processing of expectation in the social domain. This investigation will shed light on the domain-generality of the expectancy mechanism, offering an insight into whether there is a shared mechanism of detecting deviations from expectancy both in lower-level cognitive domains (music), and in higher-level domains (social decision-making). Moreover, it will further our understanding on how low-order cognitive processes such as those involved in music perception may influence complex high-order processes such as social decision-making [22].

The interdisciplinary investigation of social norms employs neurocognitive methodology and paradigms borrowed from Game Theory, such as the Ultimatum Game [20], a widely employed paradigm to investigate fairness perception. Findings using this task show that people typically opt for fair divisions of resources, even when this goes against their self-interest. In the Ultimatum Game, for example, people prefer to reject unfair deals rather than accept a certain sum of money, indicating that the willingness to react to unfairness, i.e. the unexpected outcome, is stronger than monetary incentives (what Fehr & Schmidt define inequality aversion [23]). Interestingly, if expectations are manipulated beforehand by saying that, on average, proposers tend to be unfair (or fair), the chances of rejecting unfairness diminish (or increase) [24]. These findings not only show that manipulating expectations leads to behavioural change, but also suggest that there is a default expectation, and it is related to equality (50/50 share): when there is no other reason to expect otherwise (e.g. merit or need), people, on average, identify fairness with equality and thus expect an equal allocation of resources [25,26]. The more the allocation deviates from the expected equal outcome, the stronger the reaction becomes, both behaviourally and neurally: specifically, the anterior insula (AIns) and the anterior cingulate cortex (ACC) track the gradient of this deviation from the expected equal outcome [3,27,28], irrespective of the personal advantage or disadvantage for the participant [27,28]. Reaction times (RTs) findings support the idea that deciding upon unfair offers requires more deliberation, with people being slower when accepting or rejecting unfair compared to fair offers in an Ultimatum Game [29,30]. It is worth noting, however, that some studies find that the difference in RTs is maximum when comparing fair to mid-value unfair offers (e.g. £3 out of £10) [31], suggesting that the cognitive effort is highest when the offer is ambiguous, i.e. not extremely unfair.

Electrophysiological evidence has shown that receiving unfair offers in the Ultimatum Game elicits a medial frontal negativity (MFN) in the EEG [31,32], which is modulated by participants concerns for fairness [32]. These results (i.e. frontal negativity associated with unfairness) have been replicated using a third-party Dictator Game with altruistic punishment, in which participants were given the chance to punish someone who behaved unfairly [33]. Moreover, in line with the neuroimaging literature cited above [3,27,28], some studies found that MFN is also elicited by advantageous inequality, supporting the idea that this EEG component represents fairness sensitivity rather than self-centred valence [34,35]. Importantly, this negative EEG component originates in the ACC [36], which is, as explained above, a key brain area in signalling unfairness and expectation violation.

Following these considerations and the observations of distinct behavioural and neural responses to the violation of expectations in each domain, the current study asks whether the presence of unexpected music will influence the way unexpected (i.e. unfair) social stimuli are processed. To do so, we propose to embed a manipulation of melodic expectation within a social decision-making paradigm, whereby musically expected and unexpected stimuli will be simultaneously presented with fair and unfair divisions in the social domain. We will use a computerized one-shot multi-trial third-party punishment paradigm, similar to [26]: participants will observe two players equally sharing a sum of money until one (offender) steals from the other (victim). Participants will have the chance to use some of their own monetary endowment, received before the experiment, to punish the offender. We will measure both behavioural and EEG responses in the third-party punishment task by measuring: (i) the rate of punishment; (ii) the amount spent to punish; (iii) the RTs of the choice (punishment or no punishment); and (iv) the amplitude of the MFN.

If violations of expectations in music influence the processing of fairness in the social domain, we expect that both behavioural and EEG responses will change as a function of unfairness and melodic expectancy, which will either enhance or reduce sensitivity to unfairness, as follows.

(i) If detecting a violation of expectation in music enhances the sensitivity to unfairness, we should observe an increase in punishing choices and amount spent, faster RTs and a bigger MFN amplitude in response to unfairness when exposed to unexpected compared to expected music.

This result would suggest the existence of a shared expectancy mechanism whose activation in one domain facilitates the expectancy process in a second domain (priming).

(ii) If processing unexpected stimuli in one domain (music) reduces the sensitivity to unexpected stimuli in the other domain (i.e. unfairness in the social domain), we should observe a decrease in punishing choices and amount spent, longer RTs and a smaller MFN amplitude in response to unfairness when exposed to unexpected compared to expected music. This result would suggest the existence of a shared, but limited, pool of expectancy resources, whose depletion in one domain reduces the availability in a second domain (resource competition).

# 2. Methods

## 2.1. Participants

Based on power calculations detailed below, 60 participants were planned to be recruited for this study.

The size of the interaction between music and decision-making is our effect of interest. The effect size observed in similar paradigms, conditions and populations and using similar statistical tests is in the range of $\eta_P^2 = 0.20 - 0.26$ [15,37]. Nevertheless, a more conservative medium size effect was chosen to calculate our sample. A power analysis run in G*Power [38] using a medium effect size ($\eta_P^2 = 0.06$) shows that 60 participants are needed to reach a power of 0.80 when running a repeated measure design. Considering that the estimated effect size is lower than the one suggested in the literature for similar paradigms, we are confident that our chosen sample size is adequate to address the experimental question.

In total, 67 participants were recruited because seven participants had to be removed for a variety of reasons: there were EEG recording errors for participants 2, 3, 21, 49 and 58, whose data had missing trials; the EEG data for participant 7 was accidentally not saved on disc; participant 63 had incomplete behavioural data recording. In total, we retained the data for 60 participants, as initially planned. Participants mean age was 25.6 (s.d. = 8.6), with 13 males. Participants were recruited at London South Bank University among students and staff using flyers distributed on campus and via the Research Participation Scheme system within the Division of Psychology. Exclusion criteria included: participants with neurological problems, minors, previous head injuries, self-reported hearing problems, extremely irritable skin. Left-handed people and musicians were excluded in line with previous research in the music-language literature. Participants were compensated £10 or 12 research participation credits for their time, plus £5, which they were given to take part in the game and could keep at the end of the study.

The study has been approved by the Ethics Committee of the School of Applied Sciences at London South Bank University.

## 2.2. Design

Four general linear mixed models were used to analyse how the *melodic expectancy* (information content) and *fairness of the division* predict the behavioural and neural responses in each trial per each participant: a generalized linear mixed model was used to predict the choice (categorical data), while three linear mixed models were used to predict the amount, the RTs and the MFN amplitude (continuous data). A covariate considering participants' initial fairness expectation was added to the models.

## 2.3. Materials

### 2.3.1. Decision task and stimuli

A computerized one-shot multi-trial third-party punishment paradigm, based on [39,40], was employed (see appendix A for full instructions). Three players (A, B and C) start each trial with 200 chips each (monetary-equivalent units, 1 chip = 1 cent; 200 chips = £2). Then, player A (offender) may or may not steal chips away from player B (victim). Player A can take 0, 25, 50, 75 or 100 chips from player B. Participants are always assigned the role of player C (observer), who witnesses A's actions. After seeing player A's decision, participants must decide whether or not to intervene by spending up to 100 chips (£1) of their own monetary endowment (200 chips), received at the beginning of each trial, to punish the offender by reducing their final pay-off. If participants decide to punish, they are required to indicate how much they want to spend (from 10 to 100 chips) to implement punishment.

For each 10 chips spent by the participant, A will lose 30 chips. Participants can also decide not to intervene and walk away with 200 chips. Importantly, participants can never gain more chips than what they initially have; they can only spend to make player A poorer. Participants are told that one trial will be selected at the end to determine each player's final pay-off. Participants will witness several pairs of offenders and victims (150), with an equal allocation of trials for each of the five divisions (0, 25, 75, 50 and 100), where the offenders can steal up to half of the victims' endowment, as specified above. Participants are required to indicate their response (take, i.e. punish, or leave, i.e. not intervene) by pressing the corresponding button on the keyboard. The words 'punishment', 'offender', 'victim', 'steal' will never be mentioned in order to avoid biasing participants. A small amount of deception is involved: participants are told that A–B couples are people who played before, and that current decisions will actually influence the final pay-off of all the players involved. In reality, the divisions are pre-programmed, and the participant's final payment is fixed. The deception is necessary in order to control for the amount of trials per condition seen by each participant; to make up for this deception, participants will be debriefed, and the final fixed payment will always be higher than what they would expect if an actual random trial had been selected (fixed: £5; random trial: max £2).

It is important to note that, because no reason is given for deviating from the equal allocation, in this case fairness is identified with equality. Previous findings show that people are willing to react to unfairness by spending their own money to punish offenders, even when they are not directly involved in the unfair deal [40–43]. Moreover, the willingness to punish, and the amount spent to punish increase with the increase of unfairness. In the context of the present study, the willingness to punish the offender is considered a behavioural measure of sensitivity to unfairness, i.e. sensitivity to violations of social expectations; the amount spent to punish is considered a measure of the intensity of this sensitivity [40]. A third-party paradigm has been preferred to a first-party task, where people are directly involved in the unfair situation, because the third-party paradigm allows disentanglement of the reactions elicited by the direct personal involvement from those elicited by unfairness itself [43].

As suggested by [44], beliefs about the descriptive norm will be assessed at the beginning of the task. Specifically, participants' expectation of the typical behaviour of player A will be recorded by presenting the following question: 'Player A is fully aware of the rules of the game and knows about your role as player C (observer). How much do you think A will take from B? Select one of the following options: 0, 25, 50, 75, 100'. This information will be used in the analyses to control for participants' actual expectations of fairness.

### 2.3.2. Music stimuli

The musical stimuli used in this study are isochronous five-note melodic phrases that end with either an expected or unexpected note, and have been previously used in [15]. The melodic stimuli were created using a computational model developed by Pearce [45,46], which is a variable-order $n$-gram model which estimates the probability of the pitch of a note, given the preceding notes in the melody. The expectedness of the final notes may be expressed in units of information content, which may be thought of as the degree of unexpectedness of the note given a certain preceding context [47]. High-probability notes have low information content and are expected, while low-probability notes have high information content and are unexpected. The average information content for the unexpected notes was higher (11.85) than for the expected notes (1.84) The model parameters used here are exactly the same as those used in [7] and the model has been shown to predict listeners' melodic expectations such that high-probability notes are perceived as expected and low-probability ones as unexpected [7,8].

Each of the five-note melodies was paired with five words visually presented on screen. Each note was simultaneously presented with a word, with the final (fifth) note being either expected or unexpected. The five words were: word 1: 'A' word 2: 'takes' word 3: 'from' word 4: 'B', word 5: number indicating how much A takes from B. This number can be: 0, 25, 50, 75 and 100, with 0 being fair divisions and the rest being unfair (figure 1a). The participant was instructed to press one of two buttons depending on whether they want to punish A, or leave (i.e. not intervene). When punishment is selected, the participant is asked to indicate how much they want to spend to implement their choice (i.e. how much they want to punish; figure 1b). There were 15 trials per unfairness condition (0, 25, 50, 75 and 100), paired once with expected and once with unexpected melodies (a total of 30 trials per division). This pairing of melodies with each fairness division took into account the information content of the melodies to ensure that there was no difference in melodic expectation

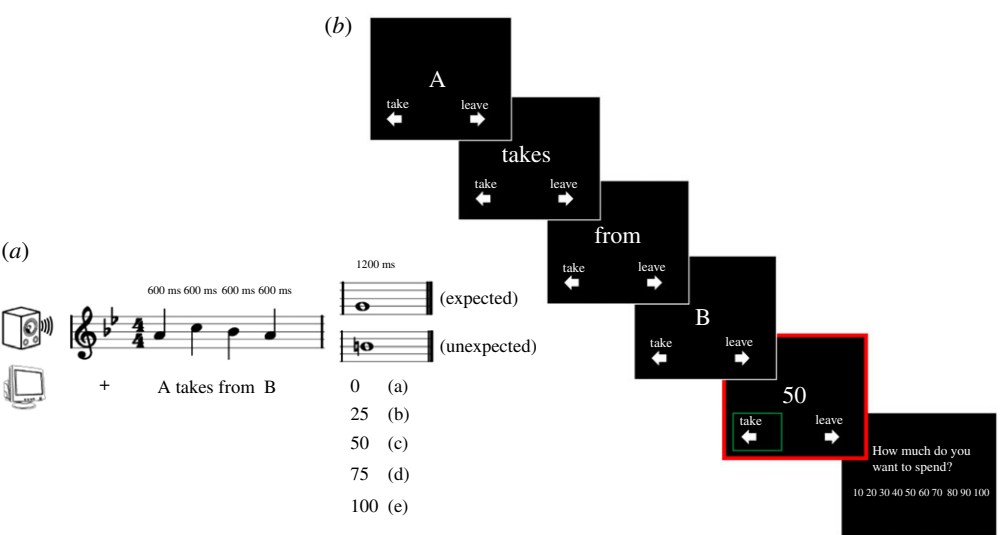

**Figure 1.** Illustration of the experimental paradigm. (*a*) Five-note melodies presented in synch with five words for the third-party punishment game. Each final note is either expected or unexpected and can be paired with either a fair division (a), or an unfair division (from b to e). (*b*) Example of a trial where the participant chooses to punish unfairness. As soon as participants see the division, they are asked to decide whether they want to punish (TAKE) or not react (LEAVE). If participants choose to punish, they will be asked to indicate the amount they want to spend on the punishment. The presentation of the division (highlighted in red) indicates the time window of interest for the analysis.

across divisions. This was also confirmed statistically ($p = 0.932$). A total of 150 trials was administered. All trials were fully randomized.

The presentation time for each of the first four word-notes was 600 ms, and the final note-word presentation was 1200 ms. Within each trial, words and notes were presented simultaneously with no breaks between them. Each trial started with a fixation cross, which stayed on screen for 600 ms. Words were presented at the centre of the screen, one word after the other, and no blank screen was presented between two words. Stimuli were presented on black background, and letters in white. The font used for the words was Arial with font size 40. Melodies were presented via earphones and the volume was kept constant across participants and for the duration of the experiment. E-prime E-studio 3 was used to present the stimuli.

## 2.4. Procedure

Participants were emailed the information about the study in advance to inform them about the procedure of the experiment and the EEG preparation. On the day of the testing, participants were welcomed in the lobby and taken to the Psychology laboratories area. Once there, they were asked to read the information again and given an opportunity to provide informed consent. After that, the EEG recording preparation began. The ActiveTwo BioSemi system (BioSemi B.V., Amsterdam, The Netherlands) was used to record continuous EEG signals from 64 electrodes using the 10/20 system. Electrodes were placed on the participant's scalp surface using an elastic electrode cap and four electrodes were placed on the participant's face to monitor eye movements: one above and one below the left eye, one on each temple. The ground electrode during acquisition was Biosemi's own Common Mode Sense active electrode and the Driven Right Leg passive electrode. Depending on the quality of the signal, this procedure took about 20 min. The participant was then asked to sit in front of a computer screen and to relax, without moving, for 2 min (1 min with eyes closed and 1 min with eyes open). Following this, participants were asked to read the instructions (appendix A) for the decision-making task, which took around 10 min. After reading the instructions, participant's initial expectation of fairness was assessed as indicated above. Finally, five practice trials were administered to allow familiarization with the task, after which participants could begin with the experimental task. The task lasted a maximum of 30 min. Breaks were given every 30 trials. Upon completion (or interruption, should the participant choose to end the experiment), the cap and electrodes were removed, and participants were given a written debrief and compensation.

## 2.5. Data cleaning

Data from each participant were cleaned following the pre-registered protocol, which is reproduced as follows:

(i) for each participant, any trial with an RT that is larger or smaller than two standard deviations from the participant's average will be excluded, as well as any null responses;

(ii) EEG data will be pre-processed as indicated below. EEG data associated with the removed behavioural data above will also be removed; and

(iii) the data for any participant who does not complete the study will not be used. Any rejected participant will be replaced in accordance with the sampling plan.

### 2.5.1. Electroencephalograpic data pre-processing

Prior to data analysis, the data were pre-processed using the standardized PREP pipeline [48]. The data epochs representing single experimental trials were extracted around the onset of the division (−1000 ms to 2000 ms, with $t = 0$ as the onset of the last note/word). Correction of artefacts were automated using MARA in EEGLAB [49,50]. The recommended rejection of independent components in MARA were checked against the guidelines provided by Pion-Tonachini *et al.* [51] and Chaumon *et al.* [52] to increase reproducibility and objectivity. The discarded independent components were recorded and saved in the data structure for each participant. Finally, the EEG data were re-segmented from −200 ms to 500 ms around the onset of the division. The EEG data were baseline corrected to 200 ms pre-stimulus period.

### 2.5.2. Data analysis

The data analysis followed the pre-registered stage 1 analysis protocol and is reported below.

Four separate linear mixed models were run to test the hypothesis of an interaction between the predictors *melodic expectancy* (information content) and *fairness of the division* on four dependent variables: the choice of punishment, the amount spent to punish, the RTs and the amplitude of MFN. A covariate considering participants' initial fairness expectation was added to the models. The R package lm4 [53] was used to run the analysis; glmer function was applied to categorical data (choice), while the lmer function was used for continuous data (amount, RTs and MFN amplitude).

For the fourth dependent variable (MFN signal), cluster-based permutation tests [54] were run. Cluster-based permutation tests are robust against the multiple comparison problem and allow us to identify the time-electrode cluster of interest without relying on pre-defined time windows and electrodes and/or visual inspection, which could lead to a bias in the statistical analysis. Cluster-based permutation tests were used to compare fair (0) versus unfair (100 division) trials over the post-stimulus (onset of the division) time window of 0–500 ms at frontal electrodes (AF3, F1, F3, F5, FC5, FC3, FC1, AF4, AFz, Fz, F2, F4, F6, FC6, FC4, FC2, FCz, Cz, C5, C6, C1, C2, C3, C4, Cz). This is based on the expectation that the MFN is frontally distributed and occurs in this time window, as identified in previous studies [30,33–36]. Following identification of the time-electrode cluster showing the maximum signal difference, signal for each trial was entered in the linear mixed effect model.

### 2.5.3. Pre-registered hypotheses

Our hypothesis of an underlying shared neurocognitive mechanism between music and social decision-making will be fully supported if an interaction is shown in both EEG data and behavioural data. This would suggest a common underlying network for processing expectation across domains. Depending on the direction of the interaction, expectation in music either enhances or reduces the sensitivity to fairness and this interaction not only affects the EEG data but also affects the behaviour in response to fairness violations.

Our hypothesis will be partially supported if either of the following outcomes are observed:

(i) an interaction is only observed in the EEG data. This would suggest that the two processes share at least an underlying neural mechanism, but this interaction is not reflected in the behavioural dependent variables considered; or

(ii) an interaction is only observed in the behavioural data. This may suggest that, despite a cognitive modulation of the behavioural outcome, the neural interaction is not observed in the event-related

data of interest and instead other components (e.g. induced activity in the time-frequency power spectrum [45]) may need to be explored.

The current analysis will not provide sufficient information to support our hypothesis if the results show an interaction in the EEG and behavioural data, but the pattern observed is in opposite directions (a reduced EEG signal with an enhanced rate of punishment, or vice-versa). This may suggest that an underlying shared mechanism exists, but a third neurocognitive mechanism may play a role. It may be that the neural interaction does not directly map onto behaviour and further studies will need to address this.

Before testing our hypothesis, we ran several pre-registered analyses to test outcome-neutral conditions.

To confirm the quality of the EEG data, we expect:

(i) an N1 component locked to the onset of the final note;
(ii) an MFN component locked to the onset of the division.

To ensure that the data are suitable to test our hypothesis, we expect:

(i) the rate of punishment for unfair divisions and the amount spent to punish to be significantly higher than rate of punishment and amount spent for fair divisions;
(ii) a larger N1 for unexpected versus expected notes;
(iii) a difference in the MFN between unfair versus fair divisions, with a larger MFN for unfair divisions.

# 3. Results

## 3.1. Planned pre-registered analyses

Behavioural and EEG data were cleaned and pre-processed as indicated in the Methods section and following the approved stage 1 protocol (available on the Open Science Framework [55] at osf.io/kwbfp). The data and code used for the analyses are available on Dryad [56] (https://dx.doi.org/10.5061/dryad.kd51c5b3r).

Four separate linear mixed models were run on our four dependent variables: (i) choice (punish or leave), (ii) RTs, (iii) amount (money spent to punish), and (iv) MFN amplitude.

The continuous predictors for the model were *melodic expectancy*, operationalized in terms of information content (the higher the information content, the more unexpected the melody) and *fairness of the division* (*fairness* from here on), operationalized in terms of money taken from the victim (the higher the value of the division, the higher the unfairness); even if fairness of the division had only five categories, we treated the variable as continuous since it is numerical and the distance between the categories is equal. Participants' initial fairness expectation, operationalized in terms of expected money taken from the victim, was added as a covariate.

For each dependent variable, we considered four models, varying the random effects: one model included only random intercepts for the between-subject predictors (participants and block); the second model included the by-participant random slopes for the within-subject predictors (*melodic expectancy* and *fairness*); the third and the forth models included the by-participant random slope for *melodic expectancy* and *fairness*, respectively. Because the variance for the random effect of block was 0, the models had an issue of singularity; for this reason, the random effect of block was dropped, and the singularity issue was solved. Based on the Akaike information criterion (AIC) index, the best model was the second one, including both by-participant random slopes, for all the dependent variables; an ANOVA comparison showed that this second model was significantly better than the others for choice and amount, although it did not differ from the third model (by-participants slope for *fairness*) for RTs and MFN. In order to test for both the main effects and interaction (*melodic expectancy × fairness*), the predictor variables were scaled using the *scale* function in R.

### 3.1.1. N1 component

We first checked the quality of the data to ensure that participants processed the final note of the melodies. A clear N1 component was elicited in response to the final note, as visible in figure 2. To ensure that participants distinguished between expected and unexpected notes, we statistically

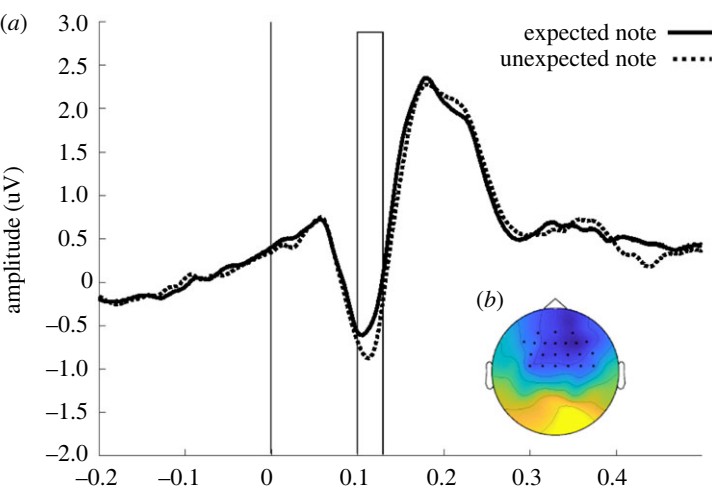

**Figure 2.** (a) Grand average ERP amplitude for expected (dashed line) and unexpected notes (solid line) at significant electrodes. The vertical line represents the onset of the note ($t = 0$). The area inside the rectangle represents the window of maximum difference of the N1 component detected by the statistical test. (b) Scalp map of the difference for the grand average ERP amplitude between unexpected and expected notes at the significant time window (104–157 ms) and significant electrodes (represented with asterisks).

compared the N1 component for expected and unexpected notes. Cluster-based permutation tests were carried out at selected frontocentral electrodes over the entire post-stimulus time window (0–500 ms). All trials were locked to the onset of the final note ($t = 0$). A significant negative cluster was found in the time window between 104 ms and 157 ms ($p = 0.007$) reflecting a decrease in the average EEG signal for unexpected versus expected notes (figure 2). This confirms the presence of a larger N1 elicited for unexpected compared to expected notes. This confirms that our data were suitable to test our hypotheses.

### 3.1.2. Choice

A generalized linear mixed model (binomial outcome variable) showed a significant effect of *fairness* (est. = 2.34, s.e. = 0.42, $z_{8530} = 5.63$ value, $p < 0.001$), which was expected, and taken as a measure of data quality assurance, and a significant effect of melodic expectancy (est. = −0.1, s.e. = 0.05, $z_{8530} = −1.91$, $p = 0.05$) on the choice; this indicates that, with a probability of 91%, participants were more likely to punish when the unfairness increased, and, with a probability of 48%, less likely to punish when the melodies were more unexpected. Nevertheless, we note that the latter effect is quite small, and the $p$-value increases ($p = 0.07$) when only main effects are considered; therefore, we will interpret this result taking this into account. No significant interaction effect was found ($p = 0.27$). Participants' initial fairness expectation (covariate) was not significant (est. = 0.46, s.e. = 0.24, $z_{8530} = 1.91$, $p = 0.06$). The magnitude (probability) and the significance of these fixed effects are plotted in figure 3a (sjplot R package [57]).

### 3.1.3. Reaction times

A general linear mixed model (continuous outcome variable) did not show any significant linear effect of the predictors of interest (all $p$-values > 0.2). The standard estimates of these fixed effects are plotted in figure 4.

### 3.1.4. Amount

A general linear mixed model (continuous outcome variable) showed a significant linear effect of fairness (est. = 12.52, s.e. = 1.75, $t_{59} = 7.14$, $p < 0.001$), indicating that, as expected, the money that participants spent to punish increased with more unfair divisions. No other effect was significant (all $p$-values > 0.1). These fixed effects, along with their standard estimates and significance are plotted in figure 5.

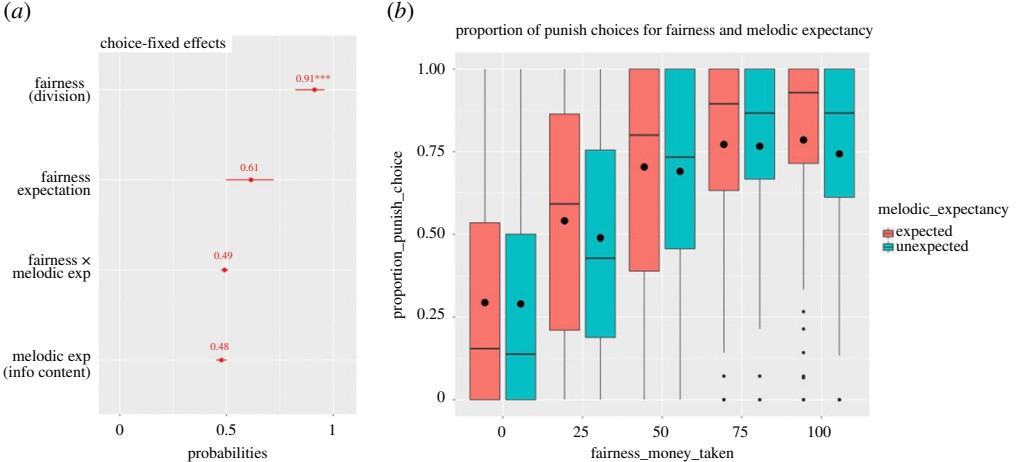

**Figure 3.** (*a*) Magnitudes (probabilities), error bars and significance (*** $p < 0.001$) of the fixed effects of the mixed model. (*b*) Boxplots for the proportion of punishing choices for all divisions presented on unexpected and unexpected notes. The black horizontal lines represent the median and the black dots represent the mean for each condition.

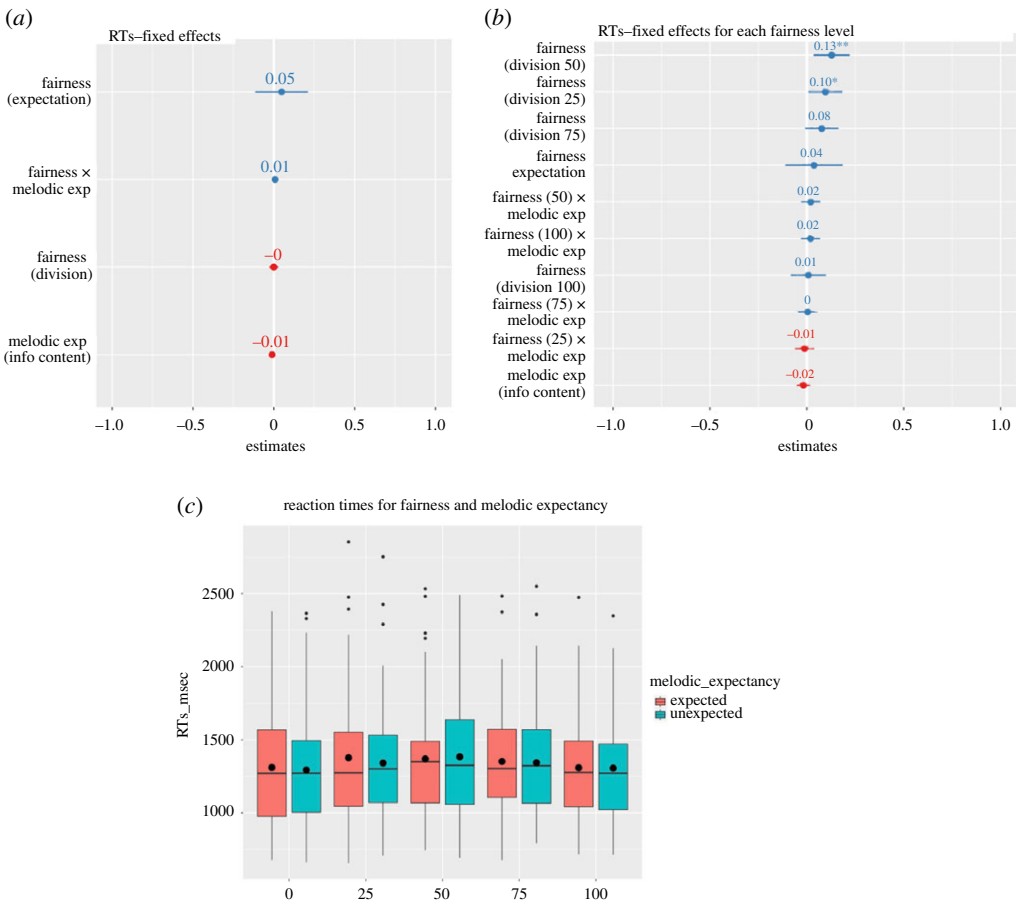

**Figure 4.** (*a*) Magnitudes (standardized estimates) and error bars of the fixed effects of the mixed model. (*b*) Magnitudes (standardized estimates), error bars and significance (* $p < 0.05$; ** $p < 0.01$) of the fixed effects of the mixed model with fairness level as factor (see exploratory analysis) (*c*) Boxplots for the RTs of the choice for all divisions presented on unexpected and unexpected notes. The black horizontal lines represent the median and the black dots represent the mean for each condition.

### 3.1.5. Medial frontal negativity

As specified in the Methods section, we used a data-driven approach to investigate whether unfair (100) divisions elicited a larger negativity compared to fair (0) divisions. Cluster-based permutation tests were

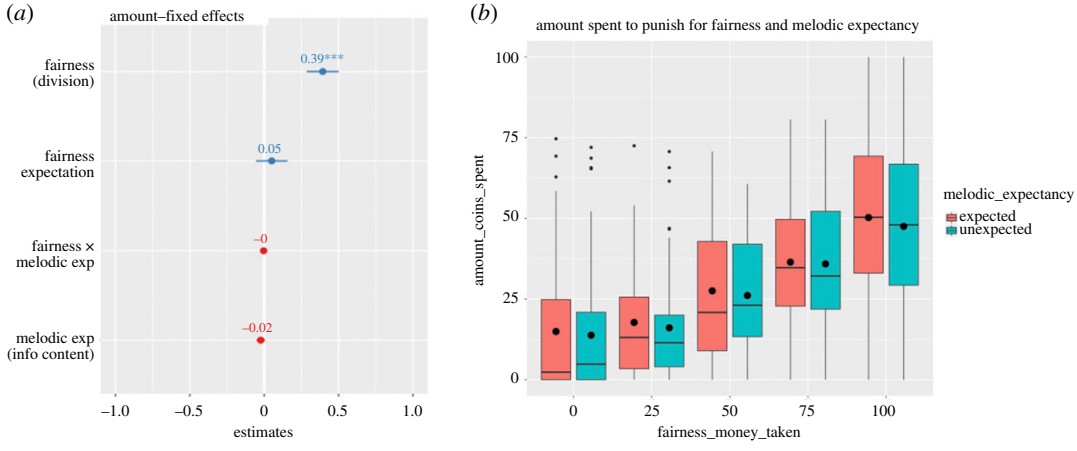

**Figure 5.** (a) Magnitudes (standardized estimates), error bars and significance (*** $p < 0.001$) of the fixed effects of the mixed model. (b) Boxplots for the amount spent to punish for all divisions presented on unexpected and unexpected notes. The black horizontal lines represent the median and the black dots represent the mean for each condition.

used to compare trials presented on unfair (100) and fair (0) divisions, averaged across melodic expectancy. The analysis was carried at all frontocentral electrodes (AF3, F1, F3, F5, FC5, FC3, FC1, AF4, AFz, Fz, F2, F4, F6, FC6, FC4, FC2, FCz, Cz, C5, C6, C1, C2, C3, C4, Cz) over the entire post-stimulus time window (0–500 ms). All trials were locked to the onset of the final note ($t = 0$). A significant negative cluster was observed at 234–268 ms over left frontocentral electrodes ($p = 0.01$), indicating a larger negative deflection for unfair versus fair divisions. A similar but not significant ($p = 0.06$) negative cluster was also observed at 287–299 ms over the same electrodes. This effect, illustrated in figure 6a, confirms a decrease in the EEG amplitude for unfair (100) versus fair (0) divisions, in line with the MFN component, which was expected and taken as a measure of data quality assurance. From visual inspection, the two significant clusters appear to be part of the same effect and were therefore combined into one cluster for subsequent single-trial analyses (time: 234–299 ms; significant electrodes in figure 6b).

In order to test for the interaction between fairness and melodic expectancy, we carried out a single-trial analysis. The EEG amplitude (μV) was extracted and averaged at each trial at the significant time-electrode MFN cluster. The average amplitude of this MFN cluster was then fed into a general linear mixed model, revealing a significant linear effect of fairness of the division (est. = −0.16, s.e. = 0.06, $t_{58.94} = -2.7$, $p = 0.009$), indicating that there was a larger decrease in the EEG amplitude (negativity) for the MFN cluster as unfairness increased. No other effects were significant (all $p$-values > 0.3). The standard estimates and significance of these fixed effects are plotted in figure 6c.

We also performed a more traditional ERP analysis on the current data, which was not included in our pre-registered analysis plan. The inherently high trial-by-trial variability and high signal-to-noise ratio (SNR) in the EEG signal could have made it difficult to detect the signal of interest using a single-trial analysis without further processing [58–60]. Therefore, we have used the most common and traditional method to increase the SNR, which is by averaging across trials and obtaining the ERP response. We averaged the EEG signal across trials to obtain the ERP response for each condition at the significant MFN cluster reported above. In order to carry out a factorial analysis, we referred back to the initial classification of the melodies as done in [11] into either expected or unexpected (see Methods for details). A 2 × 5 repeated measures ANOVA was performed in SPSS with factors melodic expectancy (expected, unexpected) and fairness (0, 25, 50, 75, 100) on the mean ERP amplitude (μV). The Greenhouse Geisser correction was used because Mauchly's test indicates a violation of sphericity for the effect of fairness and for the interaction (fairness: $\chi^2_9 = 43.02$, $p < 0.001$; interaction: $\chi^2_9 = 54.79$, $p < 0.001$). This analysis confirmed a significant effect of fairness ($F_{2.99,176.36} = 3.37$, $p = 0.02$, $\eta^2_p = 0.054$), but no significant effect of melodic expectancy ($F_{1,59} = 2.30$, $p = 0.135$, $\eta^2_p = 0.037$) and no significant interaction ($F_{2.59,152.85} = 0.475$, $p = 0.67$, $\eta^2_p = 0.008$). The effects are plotted in figure 6d. We also note that in cases where there is a serious violation of sphericity, multivariate tests are recommended, and specifically Pillai's Trace could be considered as it is robust to violations of sphericity [61]. Therefore, for completeness, we report here the results of Pillai's Trace for the effect of fairness ($F_{4,56} = 2.95$, $p = 0.028$, $\eta^2_p = 0.178$) and for the interaction ($F_{4,56} = 0.41$, $p = 0.80$, $\eta^2_p = 0.028$).

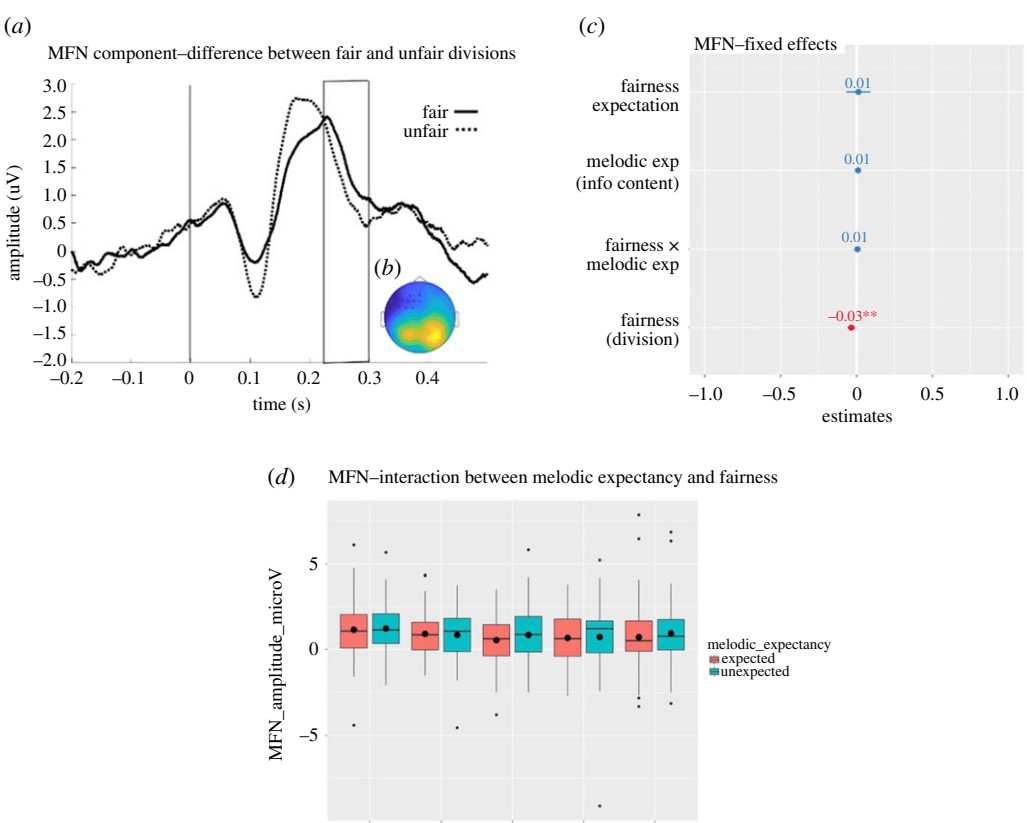

**Figure 6.** (*a*) Grand average ERP amplitude for fair (solid line) and unfair (dashed line) divisions at significant electrodes. The vertical line indicates the onset of the presentation of the division. The area inside the rectangle represents the window of maximum difference of the MFN component detected by the statistical test. (*b*) Scalp map of the difference for the average ERP amplitude between unfair and fair divisions at the significant time window (234–299 ms) and significant electrodes (represented by asterisks). (*c*) Magnitudes (standardized estimates), error bars and significance (** $p < 0.01$) of the fixed effects of the mixed model. (*d*) Boxplots for the grand average ERP amplitude at the significant MFN cluster for all divisions presented on unexpected and unexpected notes. The black horizontal lines represent the median and the black dots represent the mean for each condition.

## 3.2. Exploratory analyses

Based on the literature, we decided to further zoom in on the MFN effect by specifically looking at mid-value divisions (25, 50, 75). These have been shown to 'stand out' compared to fair and unfair divisions. For example, participants are slower in reacting to mid-value divisions [29,31], and a larger and delayed negativity (N350) has been observed for mid-value offers compared to fair and extremely unfair offers in an Ultimatum Game [31]. These results have been thought to reflect the increased difficulty of processing a more ambiguous stimulus. Therefore, our study extended the initially planned investigation to test whether mid-value divisions were processed differently from fair and extremely unfair offers in both EEG and behavioural data. Also, we extended this investigation to the interaction effect, namely, to explore whether melodic expectancy influences more ambiguous violations of fairness represented by mid-value divisions.

### 3.2.1. Mid-value divisions: reaction times

We first tested RTs by fitting a second model considering fairness as a 5-level factor, with division = 0 (fair division) as reference. This analysis showed that, compared to when the division was 0 ($M = 1301.83$; s.e. = 52.73), there was a difference in RTs when the division was 25 ($M = 1358.88$; s.e. = 56.39; est. = 57.94, s.e. = 15.81, $t_{2344.63} = 3.66$, $p < 0.001$), 50 ($M = 1376.22$; s.e. = 52.02; est. = 75.43, s.e. = 17.96, $t_{305.32} = 4.2$, $p < 0.001$) and 75 ($M = 1347.31$; s.e. = 49.76; est. = 46.89, s.e. = 21, $t_{114.59} = 2.23$, $p = 0.03$). No difference was found for unfair divisions ($M = 1307.21$; s.e. = 47.58; $p = 0.8$). This means that, compared

to fair divisions, participants were slower when deciding for mid-value divisions, but not for unfair, suggesting that mid-value divisions, especially 50, are more difficult. However, no interaction with melodic expectancy was found (figure 4).

### 3.2.2. Mid-value divisions: late medial frontal negativity

In line with our data-driven approach used in the pre-registered analysis, we first investigated whether mid-value divisions (50) significantly differed from fair (0) and unfair (100) divisions. Therefore, separate cluster-based permutation tests were carried out to compare mid-value (50) versus unfair (100) divisions, and mid-value (50) versus fair (0) divisions. These tests were carried out for all frontocentral electrodes (figure 1) over the entire post-stimulus time window (0–500 ms), locked to the onset of the division ($t = 0$). When comparing mid-value and fair divisions, cluster-based permutation tests revealed a significant negative cluster at 227–389 ms at frontocentral electrodes ($p < 0.001$) (figure 7b), indicating a decrease in EEG amplitude for mid-value versus fair divisions. When comparing mid-value and unfair divisions, cluster-based permutation tests revealed a negative cluster at 250–383 ms in the same frontocentral electrodes as above ($p < 0.001$) (figure 7c), indicating a decrease in EEG amplitude for mid-value versus unfair divisions. Altogether, these results confirm the presence of an MFN effect for mid-value divisions (50) compared to fair (0) and unfair (100) divisions. Because it is spatio-temporally similar to the MFN reported earlier, but it extends to a later time window (figure 7a), we will refer to it as the late-MFN effect.

To obtain the signal for the single-trial analysis, the EEG amplitude for each trial was extracted and averaged at the significant time-electrode cluster that encompasses both late-MFN effects (227–289 ms at significant electrodes). The average amplitude (µV) of this late-MFN cluster was fed into a linear model considering fairness as a 5-level factor, with division = 0 (fair division) as reference. This analysis showed that, compared to when the division was 0 ($M = 1.08$; s.e. = 0.16), there was a difference in the EEG amplitude when the division was 25 ($M = 0.77$; s.e. = 0.12; est. = −0.30, s.e. = 0.13, $t_{6524.32} = -2.29$, $p = 0.02$), 50 ($M = 0.75$; s.e. = 0.14; est. = −0.32, s.e. = 0.13, $t_{1314.71} = -2.39$, $p = 0.02$) and 75 ($M = 0.79$; s.e. = 0.17; est. = −0.29, s.e. = 0.13, $t_{301.14} = -2.17$, $p = 0.03$). No difference was found in the EEG amplitude between unfair ($M = 0.96$; s.e. = 0.17) and fair divisions ($p = 0.4$) and no significant interaction between melodic expectancy and fairness levels was observed (all $p$-values > 0.1). These effects are plotted in figure 7d.

Considering the more traditional ERP approach used earlier, we averaged the EEG signal across trials to obtain the ERP response for each condition at the significant late-MFN cluster reported above. We then carried out an analogous 2 × 5 repeated measures ANOVA with factors melodic expectancy (expected, unexpected) and fairness (0, 25, 50, 75, 100) on the mean ERP amplitude. The Greenhouse Geisser correction was used because Mauchly's test indicates a violation of sphericity for the effect of fairness and for the interaction (fairness: $\chi^2_9 = 41.85$, $p < 0.001$; interaction: $\chi^2_9 = 63.62$, $p < 0.001$). This analysis confirmed the effect of fairness ($F_{3.02,178.32} = 2.63$, $p = 0.05$, $\eta^2_p = 0.043$; quadratic contrast: $F_{1,59} = 10.53$, $p = 0.002$, $\eta^2_p = 0.151$). However, there was neither a significant main effect of melodic expectancy ($F_{1,59} = 0.45$, $p = 0.51$, $\eta^2_p = 0.008$) nor a significant interaction ($F_{2.44,143.85} = 1.18$, $p = 0.316$, $\eta^2_p = 0.020$). The effects are plotted in figure 7e. As mentioned earlier, multivariate tests are recommended when there is a serious violation of sphericity, and Pillai's Trace can be considered. For completeness, in line with previous results, we also report Pillai's Trace for the effect of fairness ($F_{4,56} = 3.02$, $p = 0.025$, $\eta^2_p = 0.177$) and for the interaction ($F_{4,56} = 2.18$, $p = 0.083$, $\eta^2_p = 0.135$).

### 3.2.3. P2 component

Upon visual inspection of figure 6, we observed an increased positivity for trials presented with unfair compared to fair divisions in the P2 time window. To test whether this difference is significant, we used a similarly data-driven approach as used for the MFN component. Cluster-based permutation tests were carried out to compare unfair (100) with fair divisions (0), averaged across melodic expectancy, for the post-stimulus time window of 0–500 ms, locked to the onset of the division ($t = 0$). A significant positive cluster was observed between 137 ms and 215 ms at frontocentral electrodes, $p < 0.001$ (figure 8a,b). This effect confirms the presence of a P2 component, which is larger for unfair compared to fair offers.

Given the clear effect of fairness in the P2 time window, we investigated the possibility that melodic expectancy and fairness of division may interact during this earlier time window. In order to test these effects, we carried out a single-trial analysis mirroring the analyses used for the earlier effects. The EEG

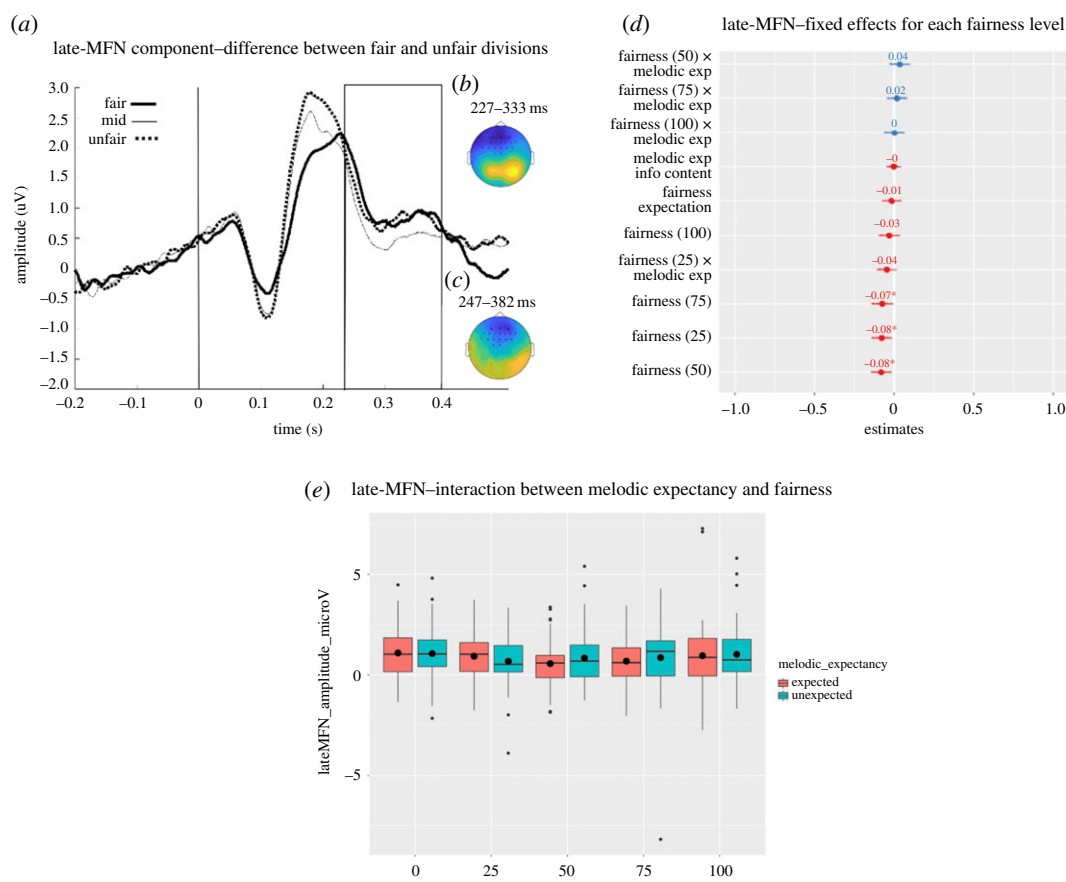

**Figure 7.** (*a*) Grand average ERP amplitude for trials presented on fair (thick solid line), unfair (dashed line), and mid-value (50) divisions (thin solid line) at significant electrodes. The vertical line indicates the onset of the presentation of the division. The area inside the rectangle represents the window of maximum difference of the late-MFN component detected by the statistical test. (*b*) Scalp map showing the difference for the average ERP amplitude between mid-value and fair divisions at the significant time window (227–333 ms). (*c*) Scalp map showing the difference for the average ERP amplitude between mid-value and unfair divisions at the significant time window (247–382 ms). Significant electrodes are represented by asterisks. (*d*) Magnitudes (standardized estimates), error bars and significance (*$p < 0.05$) of the fixed effects of the mixed model with fairness level as factor. (*e*) Boxplots for the grand average ERP amplitude at the significant late-MFN cluster for all divisions presented on unexpected and unexpected notes. The black horizontal lines represent the median and the black dots represent the mean for each condition.

amplitude (µV) was first extracted and averaged at each trial at the significant time-electrode P2 cluster. The average amplitude of this P2 cluster was fed into a general linear mixed model following the earlier analyses, thus considering the linear fixed effects of fairness and melodic expectancy. This analysis confirmed a significant linear effect of fairness (est. = 0.21, s.e. = 0.04, $t_{154.11}$ = 5.1, $p < 0.001$), indicating a larger increase (positivity) of the EEG amplitude as unfairness increased, thus confirming the P2 effect. There was also a main effect of melodic expectancy (est. = −0.09, s.e. = 0.04, $t_{494.67}$ = −2.22, $p = 0.03$), indicating that an increase in melodic expectancy (i.e. lower information content) predicted an increase in the EEG amplitude. The interaction between fairness and melodic expectancy was also significant (est. = 0.08, s.e. = 0.04, $t_{8448.42}$ = 2, $p = 0.04$), suggesting that melodic expectancy had an effect on the EEG amplitude of the P2 cluster for fairer divisions (0, 25), while this effect diminished for more unfair divisions. These effects are plotted in figure 8*c*.

Considering also the more traditional ERP approach, we averaged the EEG signal across trials to obtain the ERP response for each condition at the significant P2 cluster reported above. We carried out a 2 × 5 repeated measures ANOVA to investigate the effect of melodic expectancy (expected, unexpected) and fairness of the division (0, 25, 50, 75, 100) on the mean ERP amplitude. The Greenhouse Geisser correction was used because Mauchly's test indicates a violation of sphericity for the effect of fairness and for the interaction (fairness: $\chi^2_9$ = 32.29, $p < 0.001$; interaction: $\chi^2_9$ = 57.70,

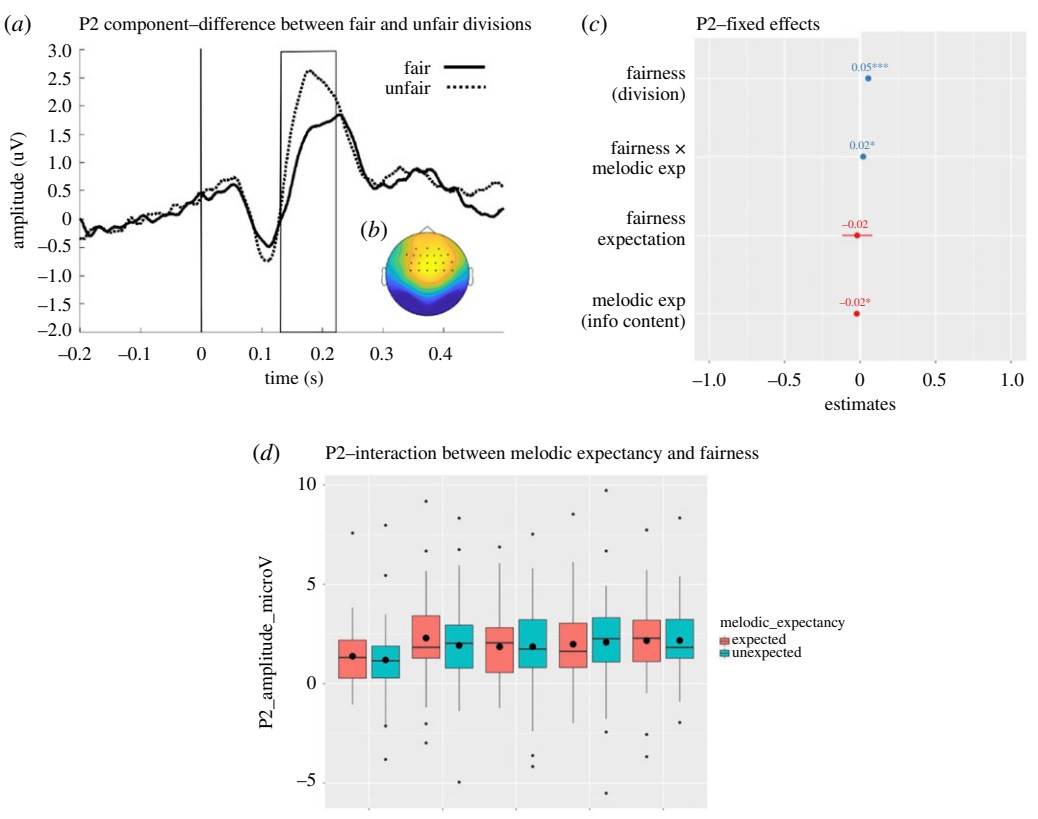

**Figure 8.** (*a*) Grand average ERP amplitude for fair (solid line) and unfair (dashed line) divisions at significant electrodes. The vertical line indicates the onset of the presentation of the division ($t = 0$). The area inside the rectangle represents the window of maximum difference of the P2 component detected by the statistical test. (*b*) Scalp map of the difference for the grand average ERP amplitude between unfair and fair divisions at the significant time window (137–215 ms) and significant electrodes (represented by asterisks). (*c*) Magnitudes (standardized estimates), error bars and significance ($^*p < 0.05$; $^{***}p < 0.001$) of the fixed effects of the mixed model. (*d*) Boxplots for the grand average ERP amplitude at the significant P2 cluster for all divisions presented on unexpected and unexpected notes. The black horizontal lines represent the median and the black dots represent the mean for each condition.

$p < 0.001$). This analysis confirmed the significant linear effect of fairness ($F_{2.99,176.49} = 16.99$, $p < 0.001$, $\eta_p^2 = 0.224$), but not the significant interaction between fairness and melodic expectancy ($F_{2.57,151.71} = 1.11$, $p = 0.342$, $\eta_p^2 = 0.018$). No significant main effect of melodic expectancy was found ($F_{1,59} = 1.34$, $p = 0.251$, $\eta_p^2 = 0.022$). For completeness, Pillai's Trace results confirm a significant effect of fairness ($F_{4,56} = 16.07$, $p < 0.001$, $\eta_p^2 = 0.534$) and reveal a significant interaction effect ($F_{4,56} = 2.63$, $p = 0.044$, $\eta_p^2 = 0.158$). However, given these inconsistent results, we recognize that the interaction effect is not particularly clear, so we will interpret it accordingly in the discussion. The effects are plotted in figure 8*d*.

## 4. Discussion

The aim of this study was to investigate whether a domain-general expectancy mechanism exists and, if so, whether the activation of this system in one domain would facilitate (priming) or reduce (resource depletion) expectancy-related processes in another domain. To do so, we created a third-party punishment task in which participants had to respond to fairness violations (social domain) that were paired with either expected or unexpected melodies (music domain). We measured participants' sensitivity to fairness by looking at willingness to punish the perpetrator of the unfairness, their RTs and the MFN, an EEG component that has been linked to detecting violations of expectations in the social domain, such as unfairness. We hypothesized that, if there is a domain-general shared expectancy mechanism between the music and social domain, both behavioural and neural responses to fairness violations will be influenced by the simultaneous presentation of melodic expectancy violations. Specifically, we hypothesized that if

this were the case, this influence would manifest itself in one of two ways: it may increase (priming) or reduce (resource depletion) participants' sensitivity to unfairness when violations of fairness norms are simultaneously presented with unexpected, compared to expected melodies.

## 4.1. Melodic and fairness perception

The data passed the initial quality checks, confirming that both our fairness and melodic expectancy manipulations work as expected. We observe a larger N1 for unexpected compared to expected melodies, and a larger MFN for unfair compared with fair divisions. Also, participants punished unfair more than fair divisions. This was expected and confirmed that the data were suitable to test our hypotheses. However, it is worth noting that we observed a rate of around 30% of punishing choices for fair divisions, which is more than what was observed in other studies [40,42]. It is not clear what drives these choices: they could be mistakes (pressing the wrong button) or genuine choices to decrease player A's pay-off. If the latter is true, these choices could be either the result of a failure to perceive player A as a different person in each trial (although instructed to do so), or an actual decision to spend money to decrease A's pay-off even in the absence of any violations [62,63]; the current design does not allow further speculation on this finding.

## 4.2. Interaction between melodic and fairness violations

The present study does not provide conclusive results to show a shared cross-domain expectancy mechanism exists. When looking at the pre-planned variables, i.e. the choice to punish, the amount to punish, the RTs and the MFN amplitude, the patterns of results do not show a clear significant interaction between fairness and melodic expectancy. Importantly, the effect size of the interaction was smaller than we had predicted, both for the behavioural data (choice to punish) and for the EEG signal (MFN). When estimating our sample size, we considered a medium effect size of $\eta_p^2 = 0.06$ for the interaction; this estimation considered previous studies using the same melodic stimuli to investigate interactions with language [15,37], which showed a larger effect size ($\eta_p^2 = 0.2$), and we presumed that the social and the music domains share fewer similarities than those shared by music and language. Nevertheless, the estimate of a medium-sized effect was still too optimistic; in fact, the current effect size for the interaction is between $\eta_p^2 = 0.02 - 0.05$, depending on whether we consider all levels of fairness or only 'extreme' levels, i.e. 0 (fair), 100 (extremely unfair) and 50 (the most ambiguous mid-value); in any case, these effects are too small to be significantly detected with our current sample size. It is nevertheless interesting that we find a small significant effect of melodic expectancy on the choice to punish fairness violations. This shows that participants are less likely to react (punish) as melodies become more unexpected (as information content increases). Owing to the absence of a significant interaction and also of any meaningful effects in the pre-planned analyses, we cannot speculate on the presence of a shared mechanism for the processing of expectations in the two domains.

Some interesting insights are derived from an exploration of mid-value divisions (25, 50, 75). In line with previous research [29,31], our behavioural and EEG data confirmed that mid-value divisions are processed differently compared to fair and unfair (100) divisions. Participants were slowest in reacting to mid-value (50) divisions, which also showed a larger negativity (late MFN), compared to fair (0) and unfair (100) divisions. This late-MFN component was more pronounced (deeper) and of a longer duration than the MFN observed for unfair versus fair divisions. Indeed, the EEG data allow us to refine our understanding of the processes underpinning this effect: the quadratic effect shows how the amplitude of this negativity may be directly linked to the ambiguity of the violation of fairness: the negativity is largest at mid-value (50) divisions and gradually reduces as the violation of fairness becomes less and less ambiguous, i.e. as it approaches the extremes of absolute fairness (0) and absolute unfairness (100). We believe that the longer processing times and larger negativity reflect the increased difficulty of processing the degree of unfairness, from most ambiguous to least ambiguous [31]. From visual inspection of the results, the effect of melodic expectancy varies with different levels of fairness of the division. However, because the results for the interaction are not statistically significant, we believe it would not be wise to speculate further.

## 4.3. Expectancy and saliency of fairness and melodic stimuli (P2)

Previous findings suggest that a positive ERP component (P2), thought to reflect attentional selection, is often observed together with, or in close proximity of, the negative component (MFN), with which it

shares the same scalp topography [64]. The current literature on economic decision-making finds this component to be related to expectancy of the outcome [65,66]. Other studies specifically investigating fairness in economic games also seems to find a P2-like component paired with the MFN, although it is not always discussed as such [31,67]. In the current set of results, we observed an increased positivity as unfairness increased, and this was reflected in both the single-trial analyses and the ERP analyses, indicating the robustness of this effect. This is in line with the interpretation that unfairness represents the unexpected outcome. Interestingly, our single-trial analysis also showed a significant effect of melodic expectancy and a significant interaction. These show that the P2 amplitude increased as information content decreased (i.e. as melodies were more expected), and that this effect was reduced for more unfair divisions. Upon visual inspection of the results, both the effect of melodic expectancy and the interaction seem to be driven by the fair (0) and 'mildly' unfair (25) divisions. Indeed, the P2 amplitude for more unfair divisions (50, 75, 100) does not seem to be influenced by changes in melodic expectancy. It is important to note that, however, we did not find a statistical difference in the P2 time window between expected and unexpected melodies when using our data-driven approach. This was also the case when using a traditional ERP analysis, which also did not reveal a significant interaction, despite the added benefit of the increased SNR. It is possible that our single-trial analysis was more sensitive to these P2 effects than the more traditional ERP analysis, but because of these discrepancies we must be cautious in how we interpret these results. Additionally, when looking at the effect of melodic expectancy, a previous study using the same stimuli did not observe a P2 difference between expected and unexpected melodies [17], and to our knowledge, previous studies have not consistently reported P2 effects for melodic expectancy violations (though see [68]). We find it interesting that the effects seen in the single-trial analyses are driven by the fair (0) and mildly unfair (25) divisions, with the mildly unfair (25) division clearly standing out both in comparison to the other divisions and in being the most affected by the change in melodic expectancy; however, because of these inconclusive results, we do not think it is appropriate to provide further speculation. The only clear result is the effect of fairness on the P2 amplitude, which, in previous literature, was interpreted as an effect driven by the expectancy of the stimuli (i.e. larger P2 for unexpected outcomes). However, because there is no effect of melodic expectancy on the P2 amplitude, the fairness effect must be related to saliency rather than just expectancy, meaning that unfair divisions are not only an unexpected outcome, but are also more salient and, therefore, worthy of attentional resources. This seems also in line with the previous literature on economic games where P2 is larger for outcomes that affect the participants rather than other players [69,70].

To conclude, the current study does not show clear evidence in support of the existence of a domain-general expectancy mechanism. Unfortunately, the estimation of the size of this effect was slightly too optimistic, hence our sample size failed to detect a clear significant effect. However, we believe this study has provided some useful insights for future research in the area. For example, we believe that mid-value divisions, and more generally ambiguous stimuli, are worthy of investigation in the context of the processing of expectations, particularly in the domain of social norms. Future investigations into cross-domain interactions could investigate more closely the different processes that might be affected with manipulations of the kind used in this study; it would be particularly interesting to disentangle the role played by early processes of attention allocation (e.g. P2) from later ones, such as those underpinning the MFN.

Ethics. The current study was approved by the local Ethics Committee at London South Bank University, SAS1832, in accordance with the British Psychological Society Ethics Guidelines and the LSBU Code of Practice for Research with Human Participants.

Data accessibility. Data available from the Dryad Digital Repository: https://dx.doi.org/10.5061/dryad.kd51c5b3r [56]. The stage 1 protocol along with the materials used were uploaded and time-stamped on OSF at the following link: osf.io/kwbfp.

Authors' contributions. C.C. and E.C. conceived and designed the study, analysed and interpreted the data and drafted the manuscript. R.T. collected all data. All authors gave final approval for publication.

Competing interests. We have no competing interests.

Funding. The study was funded by an internal funding scheme from the School of Applied Sciences at London South Bank University, London, UK.

Acknowledgements. The authors would like to thank the following, in alphabetical order: Ella Barry and Anna Camera for their technical support during data collection; Milan Golding for her contribution to the initial stages of the experimental set-up; Marcus Pearce for the initial creation of the melodic stimuli used in this study; Vassilis Sideropolous for the technical support with the initial stages of the EEG laboratory set-up. Finally, we also want to thank the reviewers for their important contributions to the study.

# Appendix A. Chip Division Game Instructions

The Chip Division Game is a game in which players make decisions about the distribution of chips, which corresponds to real money (100 chips = £1). You will be given 200 chips (£2) to play with for each trial of this game. At the end of the game, one trial will be randomly selected to determine your payment, that will depend on your choice on that trial.

In this game, there are three players, A, B and C. You have been randomly assigned to play in the role of C. All of the players start each trial with 200 chips (£2).

A and B played before, in previous experimental sessions, so what you are about to see are outcomes from previously taken decisions. This means that, although your choice can modify the final monetary outcome for all the players, it cannot modify the choices of the players, because these have already been made.

## A.1. Previous sessions

Players came to the laboratory and were divided in pairs. They then received 200 chips as an initial endowment. One player was randomly assigned to the role of player A and was given the chance to take chips away from player B; player B could do nothing but accept the decision of player A. Player A could decide to take 0, 25, 50, 75 or 100 chips out of the 200 chips of player B.

- (i) If player A decided to take no chips (0), then both player A and player B would end the session with an equal amount of chips/money (200 each); and
- (ii) If player A decided to take 25 chips, then player A would end the session with 225 chips, and player B with 175 chips; if A decided to take 50, then A would end with 250 and B with 150; etc.

## A.2. Current session (your session)

You will be presented with many trials: each trial corresponds to the decision of one pair of players (i.e. how many chips player A has taken from player B). Every trial represents a different pair. Each pair of players made only one decision, and you will never encounter the same pair twice.

In each trial, you can choose to do nothing and walk away (by pressing Leave, the right arrow key), or you can decide to spend some of your own chips (200 chips = £2) to take some money away from A (by pressing Take, the left arrow key). If you choose to spend money to take from A, for every chip that you spend, A is going to lose 3. So, if you decide to spend 10, A is going to lose 30; if you decide to spend 20, A is going to lose 60; etc. The minimum number of chips that you can spend in each trial is 10 and the maximum is 100, with increments of 10 chips (see picture below). This also means that player A can lose a maximum of 300 (100 × 3) chips, and this will cost you a maximum of 100 chips.

Note that the money that is taken away from A returns into the experimental pot, as none of the players will get it.

Figure 9 is an example of how a trial looks like. Each black square represents a different screen.

Each screen will appear for a very short amount of time. The last screen (with A's choice) will last for 1 s. You will need to respond by pressing Take (left arrow) or Leave (right arrow). Make sure you press the button that you intend to (in other words, be accurate), but also do not think too much and try to respond as fast as possible.

If you select 'Leave' the following screen will be displayed:

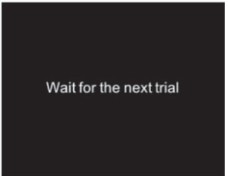

and no further response will be required.

If you select 'Take' the following screen will be displayed:

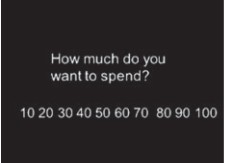

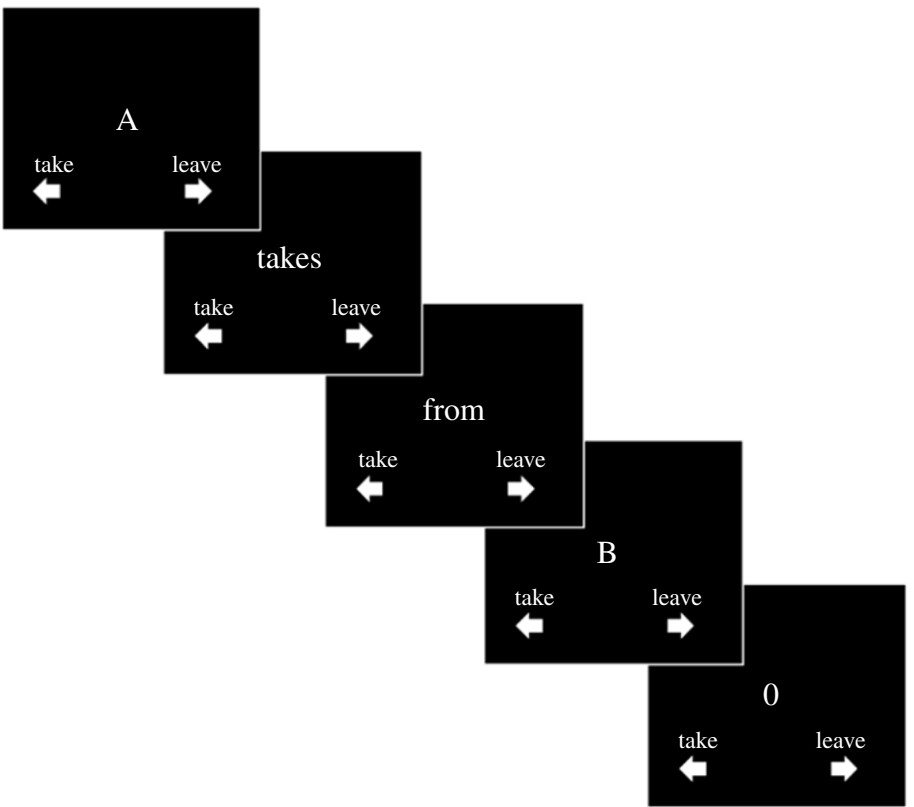

**Figure 9.** In this example trial, A decided to take 0 (no chips) from B, so they both ended the game with 200 chips each.

You will need to use the mouse to click one of the options (10, 20, 30, etc.). You will only have few seconds to decide.

## A.3. Examples

(i) If A takes 0 (no chips) from B, then both A and B have 200 chips. If you decide to Take from A and you spend 10, then you finish with 190 chips (200 − 10), player A loses 30 (10 × 3) and finishes with 170 chips (200 − 30), and player B finishes with 200 chips. If you decide to Leave, then you all finish with 200 chips.

(ii) If A takes 50 chips from B, then A has 250 chips and B has 150 chips. If you decide to Take from A, and you spend 30, then you finish with 170 chips (200 − 30), player A loses 90 (30 × 3) and finishes with 160 chips (250 − 90), and player B finishes with 150 chips. If you decide to Leave, then you finish with 200 chips, A finishes with 250 chips and B finishes with 150 chips.

At the end of the game, a random trial will be selected. If, in that trial, you spent 20 chips to Take from A, then you are going to get £1.80 (180 chips) as your additional payment. If in that trial you decided to Leave, you are going to get £2 (200 chips). Players A and B will also receive their payment according to your choice in that trial.

IMPORTANT: this is not a maths test. Don't worry if you can't make quick and accurate calculations! Nobody can! And that is not the scope of the study. Just go with your guts, and you'll be fine! There are no right or wrong answers!

Before starting the game, imaging a random pair of players in the laboratory. One player has just been selected to be player A and is told that he/she can take chips from B, if he/she want to. Please indicate how much you expect player A to take from player B.

Circle one answer: 0 (no chips)   25   50   75   100

Shortly, you will do a few practice trials. The practice trials are used only to make you familiar with the stimuli, and are not be part of the pool of real trials from which your final payment will be selected.

Do you have any questions?

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
