## [Reviewer comments · Royal Society Open Science]

Review History

Decision letter (RSOS-182045.R0)

03-Dec-2018

Dear Dr Carrus,

I write you in regards to your Stage 1 Registered Report submission RSOS-182045 entitled "Does unfairness sound wrong? A cross-domain investigation of expectations in music and social decision-making." which you submitted to Royal Society Open Science.

We routinely triage submissions for scientific soundness, clarity and general adherence to the Registered Reports guidelines. For submissions that have promise but are not yet suitable for in-depth Stage 1 review, we offer feedback to help authors maximise the chances that reviewers will respond positively to a resubmission.

We have concluded that your submission is not yet suitable for in-depth review and has therefore been rejected at this time, but we believe it will be suitable once several issues are addressed. We therefore invite a resubmission. Further comments from the Associate Editor may be found at the end of this letter.

Reports © 2020 The Reviewers; Decision Letters © 2020 The Reviewers and Editors;
Responses © 2020 The Reviewers, Editors and Authors. Published by the Royal Society under the terms of the Creative Commons Attribution License <http://creativecommons.org/licenses/by/4.0/>, which permits unrestricted use, provided the original author and source are credited

If you wish to revise your manuscript in light of the below comments please submit your manuscript as a new submission and mention this previous manuscript ID in your covering letter. You should also provide a detailed response to the below comments in the cover letter.

Thank you for considering Royal Society Open Science for the publication of your registered report.

on behalf of Professor Chris Chambers (Registered Reports Editor, Royal Society Open Science)
 openscience@royalsociety.org

Editor's Comments to the Author:

Should you decide to resubmit, please address the following issues:

1. The Introduction feels very long. Please consider whether all of the content prior to the statement of aims is necessary. If it is, I would recommend shortening paragraphs to aid readability and anticipate the aims much sooner (e.g. at the end of the first paragraph). Please also supply an abstract summarising the research question, rationale, aims and methods.
2. A clearer mapping is required between the hypotheses, power analyses, statistical test, and theoretical implications of the potential outcomes. Suggest listing presenting hypotheses in a more explicit list format at the end of the introduction and then making clear in the Analysis section (again, using a list) which hypothesis is associated with which statistical test(s), and crucially, which outcomes will confirm or disconfirm the hypothesis. You may also wish to summarise this in tabular format across all hypotheses to aid information synthesis. Every preregistered hypothesis must have its own or sampling plan (e.g. power analysis calculated analytically or through simulations) linked to the specific test. Many of your tests are not directly linked to a hypothesis or power analysis (e.g. the LME analysis). Where multiple analyses are used to assess one hypothesis, it must be crystal clear which outcomes (or combination of outcomes) would confirm or disconfirm the hypothesis in question.
3. Related to (2), contingencies for additional analyses are in many places too vague, e.g. "If our sample shows enough variability in initial expectations (as measured at the beginning of the experiment), participant's initially recorded expectations on the offender's behaviour will be considered as a covariate (2x2 repeated measures ANCOVA)." What will count as sufficient variability?
4. For added clarity, you might consider presenting the hypotheses in graphic (visual) form to make the directionality of the predicted effects clear in every case.
5. It is unclear if the sample size of 30 will be sufficient to provide a sensitive test of all hypotheses. You should justify the sampling plan according to the smallest effect size of theoretical interest, rather than using single point estimates of observed effect sizes from previous studies (which are likely biased in a positive direction). At present, reviewers are likely to consider N=30 to be insufficiently justified and thus too small given the large number of statistical tests and predictions.
6. At a higher level it is unclear what combination of behavioural and/or EEG outcomes is necessary to provide support for the two explanations (E1, E2). I would strongly recommend including an interpretative contingency plan - e.g. how will you interpret partial support (e.g. E1 or E2 supported by behavioural but not EEG data, or vice versa, and on a more fine grained scale

across the specific hypotheses). If the behavioural effects are pivotal in the interpretation, and if these contingencies across behavioural and EEG measures are complicated, might it be prudent for the Stage 1 submissions to initially focus on proposing a purely behavioural study (no EEG) in a large sample, and only once the hypotheses are confirmed to then conduct a replication with EEG added? (this latter point is only a suggestion, not a requirement)

7. One of the key criteria that reviewers are asked to assess at Stage 1 is "Whether the authors have considered sufficient outcome-neutral conditions (e.g. absence of floor or ceiling effects; positive controls) for ensuring that the results obtained are able to test the stated hypotheses", and successfully passing such tests is an editorial criterion at Stage 2 following completion of the study (see <http://rsos.royalsocietypublishing.org/registered-reports>). Please make clear which tests will serve, e.g. as positive controls for the effect of fairness and music expectedness, as well as any other outcome neutral tests to e.g. confirming quality of EEG data.

8. Exclusion criteria for data within participants, and for participants within the sample: please ensure that these are comprehensively pre-specified as it is generally not possible to adjust these for pre-registered analyses after provisional acceptance has been awarded. Please also make clear whether any excluded participants would be replaced.

9. Method of eyeblink removal in EEG data must be objectively reproducible. Subjective method (visual inspection) is insufficient unless performed by independent, blinded analysts using a clearly described reproducible methodology for which you can demonstrate high inter-rater reliability. Most RRs proposing the use of EEG use an objective algorithm to exclude eyeblinks to ensure reproducibility.

10. To ensure a clear separation between pre-registered confirmatory analyses and exploratory post hoc analyses, any mention of exploratory analysis should be removed from the Stage 1 manuscript and introduced at Stage 2 (after data collection) in the "Exploratory Analyses" section of the Results. Alternatively, if these exploratory analyses address specific hypotheses then they can be kept in the Stage 1 submission but must be fully elaborated and associated with a sampling plan (e.g. power analysis).

We hope this feedback is helpful and look forward to receiving your revised submission. Should you decide to resubmit please include a point-by-point response to the above points in your cover letter. Please note that you will be required to submit your manuscript afresh and will receive a new manuscript number.

Author's Response to Decision Letter for (RSOS-182045.R0)

See Appendix A.

RSOS-190048.R0

Review form: Reviewer 1

Is the language acceptable?

Yes

Do you have any ethical concerns with this paper?

No

Have you any concerns about statistical analyses in this paper?

No

Recommendation?

Major revision

Comments to the Author(s)

I read the stage 1 by Carrus and Civai. This is the first time I review a registered report. Thank you for the opportunity and apologies in advance if I make some mistake. My recommendation is "major revision" because I ask authors a lot of work (in particular R and matlab scripting).

The authors want to investigate with a bio measure and a behavioural measure whether a violation in one field (social) interacts with a violation in another field (music).

Hypothesis. The hypothesis is driven by that part of the current literature that suggests a relationship between music and language. I do not share much that view, but, definitely, things are out there so I have nothing to complain about the hypothesis. Perhaps, the only critic here is about how strong can be this relationship. Therefore, how strong we expect to be the relationship between the violation in music and the social violation (see below). My best guess is that this relationship is weak and small.

Method. I list here a set of critics and (when it is possible) suggestions.

The first problem seems to me the number of participants. Given that I suspect the relationship to be small, I was surprised to see that the power calculation returned 34 subjects. I even downloaded G*power and tried to calculate N myself. Unfortunately, I found parameters that I did not how to fill in. If I fill all the info reported in the paper, the software tells me "groups must be more than 2" and, in addition, there is another parameter I can't fill in because it is not reported by the authors. I'm sure the problem it's me. However, I would like authors to explain me step by step how N was calculated. I would suggest to write an R script (I will suggest this later also) but, in case, even a video or a screenshot that show the N is calculated should be fine.

I suggest also other strategies to calculate N. There is a large debate now about power calculation. Many suggest that power calculation has no sense given that data in literature may not be that reliable. Therefore, any power calculation is unreliable. A conservative approach could be that of targeting the N that enables to reveal the smallest possible effect of interest for the given effect. I explain myself. Let suppose authors think the smallest effect they would like to detect is $d=.30$ (i.e., a small effect). They should calculate N with this effect size in mind. In this way, the experiment would reveal the effect even if the effect is small. This is a very conservative approach.

In any case, 34 subjects to test the relationship between a social violation and a music violation (two very far fields) seems very small.

Another problem with the method is that materials are not available. I was curious (for example) to see the task and to listen to the melodies. I found no link to materials. Materials are just briefly described (By the way, I synthesized the score that is reported in the appendix and played the melody. Well, perhaps I have very a broad music taste. But the violation definitely did not violate me!). In synthesis, the experiment should be available so that we can try the task. Note that there is no description about the music stimulus. Which waveform? Sample rate? Bits? Listening level? Freefield or headphones? (it looks headphones from the picture).

Analysis. Behavioural measure. It is not clear what is the criterion to remove outliers in the behavioural measure. The mean of what? The mean of the participant, of the condition? The standard deviation of what? The sd of the participant, of the condition? The best way to sort everything out is to add a sample dataset (fake, with simulated data) and an R script that, from

raw data to statistical results shows what authors intend to do. If this script included the graph too, it would be brilliant. Then, when data collection is over, authors would have simply to run the script on the real data. Script and fake-data should be available like the digital materials in the next submission. I would also comment section by section the various parts of the script (e.g. from raw data to means, removing outliers...)

Analysis. EEG. This is a problem. And I'm not an EEG expert. So I asked a colleague that works with EEG. He told me the pipeline is good and standard and I trust him as well as the authors. The problem, as usual, is when the experimenter's hand comes into play. There are two problems: reproducibility of the results and reliability of the results. Reproducibility. The colleague suggested me you should write down all the signals that will be discharged at the various stages of the analysis. And report also the stage of the pipeline together with the reason why the signal was discharged. This would guarantee the reproducibility of the result you will collect. Reporting the exclusion criteria would be fundamental here. I see you are using a matlab toolbox. My extra suggestion would be that to write a batch script of the analysis you do. The batch script should start from raw data up to final analysis and, of course, it should include outliers removal. Another option I think you could try is to have a third independent party (e.g., a colleague that is working with eeg) that will be consulted to do the outlier removal. This colleague needs to be, of course, hypothesis-free. Like for the behavioural experiment, I would add a R script that from raw data takes analysis to final results, maybe a graph.

Additional control measures. I would control for the music aptitude of the participants. This could explain why the effect works or not. First of all, I would check whether there are amusic. To check for aptitude there are two options. First use an independent test (e.g., PROMS). But it takes 20 minutes.

https://www.uibk.ac.at/psychologie/fachbereiche/pdd/personality_assessment/proms/take-the-test/mini-proms/index.html.en

In alternative use a questionnaire such as the goldsmith.
<https://www.gold.ac.uk/music-mind-brain/gold-msi/>

Again on music samples. I understand violations will be theoretical violations. Are we sure participants perceive them as violations? I mean: is there any subjective measure of the "violationness" of each melody?

This measure (the PROMS, the questionnaire or the subjective rating of violation) should be taken into account or controlled for by the statistical analysis. Perhaps an estimate of the violationness of each sample would be best. This could be easily collected with a pre-study.

In synthesis,

1. I want to understand how power was calculated (N=34 seems rather small to me) and, in particular, in a very conservative approach what N should be like.
2. I would add to the submission all the possible digital materials: stimuli, experiment, analysis scripts so that, in principle, the reader could replicate the experiment here and now, including the statistical analysis and the signal analysis.
3. I would look for a control measure that somehow check whether the music violation is perceived as a violation. If any, even a subjective evaluation of a different group of subjects would be interesting.

Apologies for this late review (I'm usually faster).

Review form: Reviewer 2

Is the language acceptable?

Yes

Do you have any ethical concerns with this paper?

No

Have you any concerns about statistical analyses in this paper?

No

Recommendation?

Major revision

Comments to the Author(s)

In this report the authors describe a research proposal with the aim to investigate whether violations in expectations of music alter sensitivity to violations in fairness and influence subsequent social behaviour. The authors base their idea on literature from music and language showing how unexpected music tones affect language processing, both semantic and syntactic, and literature in social decision-making showing how violating social norm expectations affect social behaviour. The main question they seek to answer is whether expectations in music underlie a basic mechanism of expectancy that is shared across different domains such as social decision-making. To test this, they want to examine how expectations in music influence processing of fairness (with EEG) and social behaviour (decision behaviour), using an existing third-party punishment game. Their hypotheses are exploratory with each providing potential interpretations, namely 1) resource competition and 2) priming.

Main comments:

The cross-domain approach the authors attempt to investigate is interesting and novel. However, I have some concerns about the plausibility of the proposed interpretations of the hypotheses and aim of the study. First, the rationale to study music expectations with social decision-making is not entirely clear from the proposal. The authors explain how social norms can be seen as rules and expectations which are similar to language acquisition, but they do not explain why they chose to specifically investigate the relation with social decision making or how this may be relevant for social decision-making. Second, social decision-making such as fairness is a complex behaviour that involves higher-order cognition, whereas language and music (as the authors mention) involve low-order cognitive processes. Moreover, language may not be completely unrelated to melodic tones. Claims about potential shared mechanisms between these processes and how music expectations influence such complex social behaviour needs to be taken with caution.

The authors provide two accounts to explain possible results. One alternative hypothesis would be that when expectations are incongruent, an increase in conflict-monitoring occurs and as result more attention to the target after an incongruent trial (Botvinick et al., 2001). Here, for example, an expected melodic tone and unfair behaviour of player A yields incongruency in expectations which may result in increased conflict signal, increased decision time and increased punishment. To distinguish between priming and conflict, the authors could also look at the reaction time of the decision.

Furthermore, incongruent expectations may induce mood differences and influence fairness decisions. To control for mood as an explanation, one could include a mood assessment in the task (not every trial, but on some occasions) or either before and in the breaks. This can be included as a predictor of non-interest.

The design of the study allows to test the stated hypotheses.

Comments on description of the methods, experimental procedures and analysis pipeline:

- The authors state that a small amount of deception is involved, but don't explain why this is necessary. I am not against using deception, but it would be good to explain this.
- One potential issue in the task is that the authors may end up running into a floor effect for the fair condition. In particular for comparing EEG signals, this might be an issue.
- The information content part of the music stimuli is unclear and confusing. How is the information content determined and what are the implications of this in addition to whether the final note is expected or unexpected? Do unexpected notes have different information contents? If so, this would need to be accounted for in your statistical model.
- On page 7, line 4(7) the authors talk about 4 unfair offers, though, if I understand the task and materials correctly, these are not "offers" but are different amounts that player A chooses to steal/take from player B.
- From reading the materials, it is ambiguous whether the authors intend to look at choice behaviour as a binary outcome (take / leave) or only look at the choice behaviour as a continuous outcome from 0-100.
- Procedure and analysis-pipeline seem clear and sound. For the behavioural analysis, one may consider using a linear mixed effect model instead. Mixed effects model does not require to aggregate behaviour to a mean per subject, losing valuable information of individuals' behaviour. Mixed effects models generally are more powerful as you include the inter-individual variance and intra-individual variance.
- Are all trials the same length in duration? To avoid participants to finish the task quicker by choosing "leave".
- Some suggestions for the instructions in Appendix A:
 - o The instructions of the role of the players in the game are not very clearly introduced before explaining the actions the player can take. Specifically, line 25 where it explains the actions of player A without any context of the role of player A and the game. I would suggest to include some sort of cover story as to why player A can steal from B. For example, by introducing that player A has given the possibility or power to take some of the divided chips from player B to increase player A's total amount of chips (and money). At the same time, player B cannot do anything. Then introduce player C's role.
 - o The introduction of the music part comes a bit unexpected and is quite obvious that it probably should have an effect on their behaviour. Perhaps the authors could try to better fit this in the game with a cover story so it doesn't become so obvious.
 - o The Take action becomes a bit confusing with the "How much do you want to spend" screen, as in this part it is not clear how much 30 chips spend means. I would suggest to better integrate the take action, spending of own chips and how that affects punishment (x3) in the text before showing an example of the trial.

Botvinick, M. M., Braver, T. S., Barch, D. M., Carter, C. S., and Cohen, J. D. (2001). Conflict monitoring and cognitive control. *Psychol. Rev.* 108, 624–652. doi: 10.1037//0033-295X.108.3.624

Decision letter (RSOS-190048.R0)

05-Feb-2019

Dear Dr Carrus,

The Editors assigned to your Stage 1 Registered Report ("Does unfairness sound wrong? A cross-domain investigation of expectations in music and social decision-making.") have now received comments from reviewers. We would like you to revise your paper in accordance with the referee and editors suggestions which can be found below (not including confidential reports to the Editor). Please note this decision does not guarantee eventual acceptance.

on behalf of Chris Chambers (Registered Reports Editor, Royal Society Open Science)
openscience@royalsociety.org

Associate Editor Comments to Author (Professor Chris Chambers):

Two expert reviewers have now assessed the manuscript. Both see merit in the proposal, but also raise concerns that cut across the full range of Stage 1 review criteria, from the rationale of the proposed hypotheses, to the clarity and level of detail in the methods (including the power analysis), to the justification of specific methodological decisions and the consideration of necessary controls. Please also ensure that any digital materials are available to the reviewers. Such materials will be required to be publicly archived in the event of final Stage 2 submission, but as per the comments would clearly be useful to the reviewers earlier. Given these assessments, a Major Revision is recommended.

Comments to Author:

Reviewer: 1

Comments to the Author(s)

I read the stage 1 by Carrus and Civai. This is the first time I review a registered report. Thank you for the opportunity and apologies in advance if I make some mistake. My recommendation is "major revision" because I ask authors a lot of work (in particular R and matlab scripting).

The authors want to investigate with a bio measure and a behavioural measure whether a violation in one field (social) interacts with a violation in another field (music).

Hypothesis. The hypothesis is driven by that part of the current literature that suggests a relationship between music and language. I do not share much that view, but, definitely, things are out there so I have nothing to complain about the hypothesis. Perhaps, the only critic here is about how strong can be this relationship. Therefore, how strong we expect to be the relationship between the violation in music and the social violation (see below). My best guess is that this relationship is weak and small.

Method. I list here a set of critics and (when it is possible) suggestions.

The first problem seems to me the number of participants. Given that I suspect the relationship to be small, I was surprised to see that the power calculation returned 34 subjects. I even downloaded G*power and tried to calculate N myself. Unfortunately, I found parameters that I did not how to fill in. If I fill all the info reported in the paper, the software tells me "groups must be more than 2" and, in addition, there is another parameter I can't fill in because it is not reported by the authors. I'm sure the problem it's me. However, I would like authors to explain me step by step how N was calculated. I would suggest to write an R script (I will suggest this later also) but, in case, even a video or a screenshot that show the N is calculated should be fine.

I suggest also other strategies to calculate N. There is a large debate now about power calculation. Many suggest that power calculation has no sense given that data in literature may not be that reliable. Therefore, any power calculation is unreliable. A conservative approach could be that of targeting the N that enables to reveal the smallest possible effect of interest for the given effect. I explain myself. Let suppose authors think the smallest effect they would like to detect is $d=.30$ (i.e., a small effect). They should calculate N with this effect size in mind. In this way, the experiment would reveal the effect even if the effect is small. This is a very conservative approach.

In any case, 34 subjects to test the relationship between a social violation and a music violation (two very far fields) seems very small.

Another problem with the method is that materials are not available. I was curious (for example) to see the task and to listen to the melodies. I found no link to materials. Materials are just briefly described (By the way, I synthesized the score that is reported in the appendix and played the melody. Well, perhaps I have very a broad music taste. But the violation definitely did not violate me!). In synthesis, the experiment should be available so that we can try the task. Note that there is no description about the music stimulus. Which waveform? Sample rate? Bits? Listening level? Freefield or headphones? (it looks headphones from the picture).

Analysis. Behavioural measure. It is not clear what is the criterion to remove outliers in the behavioural measure. The mean of what? The mean of the participant, of the condition? The standard deviation of what? The sd of the participant, of the condition? The best way to sort everything out is to add a sample dataset (fake, with simulated data) and an R script that, from raw data to statistical results shows what authors intend to do. If this script included the graph too, it would be brilliant. Then, when data collection is over, authors would have simply to run the script on the real data. Script and fake-data should be available like the digital materials in the next submission. I would also comment section by section the various parts of the script (e.g. from raw data to means, removing outliers...)

Analysis. EEG. This is a problem. And I'm not an EEG expert. So I asked a colleague that works with EEG. He told me the pipeline is good and standard and I trust him as well as the authors. The problem, as usual, is when the experimenter's hand come into play. There are two problems: reproducibility of the results and reliability of the results. Reproducibility. The colleague suggested me you should write down all the signals that will be discharged at the various stages

of the analysis. And report also the stage of the pipeline together with the reason why the signal was discharged. This would guarantee the reproducibility of the result you will collect. Reporting the exclusion criteria would be fundamental here. I see you are using a matlab toolbox. My extra suggestion would be that to write a batch script of the analysis you do. The batch script should start from raw data up to final analysis and, of course, it should include outliers removal.

Another option I think you could try is to have a third independent party (e.g., a colleague that is working with eeg) that will be consulted to do the outlier removal. This colleague needs to be, of course, hypothesis-free. Like for the behavioural experiment, I would add a R script that from raw data takes analysis to final results, maybe a graph.

Additional control measures. I would control for the music aptitude of the participants. This could explain why the effect works or not. First of all, I would check whether there are amusic. To check for aptitude there are two options. First use an independent test (e.g., PROMS). But it takes 20 minutes.

https://www.uibk.ac.at/psychologie/fachbereiche/pdd/personality_assessment/proms/take-the-test/mini-proms/index.html.en

In alternative use a questionnaire such as the goldsmith.

<https://www.gold.ac.uk/music-mind-brain/gold-msi/>

Again on music samples. I understand violations will be theoretical violations. Are we sure participants perceive them as violations? I mean: is there any subjective measure of the "violationness" of each melody?

This measure (the PROMS, the questionnaire or the subjective rating of violation) should be taken into account or controlled for by the statistical analysis. Perhaps an estimate of the violationness of each sample would be best. This could be easily collected with a pre-study.

In synthesis,

1. I want to understand how power was calculated (N=34 seems rather small to me) and, in particular, in a very conservative approach what N should be like.
2. I would add to the submission all the possible digital materials: stimuli, experiment, analysis scripts so that, in principle, the reader could replicate the experiment here and now, including the statistical analysis and the signal analysis.
3. I would look for a control measure that somehow check whether the music violation is perceived as a violation. If any, even a subjective evaluation of a different group of subjects would be interesting.

Apologies for this late review (I'm usually faster).

Reviewer: 2

Comments to the Author(s)

In this report the authors describe a research proposal with the aim to investigate whether violations in expectations of music alter sensitivity to violations in fairness and influence subsequent social behaviour. The authors base their idea on literature from music and language showing how unexpected music tones affect language processing, both semantic and syntactic, and literature in social decision-making showing how violating social norm expectations affect social behaviour. The main question they seek to answer is whether expectations in music underlie a basic mechanism of expectancy that is shared across different domains such as social decision-making. To test this, they want to examine how expectations in music influence processing of fairness (with EEG) and social behaviour (decision behaviour), using an existing third-party punishment game. Their hypotheses are exploratory with each providing potential interpretations, namely 1) resource competition and 2) priming.

Main comments:

The cross-domain approach the authors attempt to investigate is interesting and novel. However, I have some concerns about the plausibility of the proposed interpretations of the hypotheses and aim of the study. First, the rationale to study music expectations with social decision-making is not entirely clear from the proposal. The authors explain how social norms can be seen as rules and expectations which are similar to language acquisition, but they do not explain why they chose to specifically investigate the relation with social decision making or how this may be relevant for social decision-making. Second, social decision-making such as fairness is a complex behaviour that involves higher-order cognition, whereas language and music (as the authors mention) involve low-order cognitive processes. Moreover, language may not be completely unrelated to melodic tones. Claims about potential shared mechanisms between these processes and how music expectations influence such complex social behaviour needs to be taken with caution.

The authors provide two accounts to explain possible results. One alternative hypothesis would be that when expectations are incongruent, an increase in conflict-monitoring occurs and as result more attention to the target after an incongruent trial (Botvinick et al., 2001). Here, for example, an expected melodic tone and unfair behaviour of player A yields incongruency in expectations which may result in increased conflict signal, increased decision time and increased punishment. To distinguish between priming and conflict, the authors could also look at the reaction time of the decision.

Furthermore, incongruent expectations may induce mood differences and influence fairness decisions. To control for mood as an explanation, one could include a mood assessment in the task (not every trial, but on some occasions) or either before and in the breaks. This can be included as a predictor of non-interest.

The design of the study allows to test the stated hypotheses.

Comments on description of the methods, experimental procedures and analysis pipeline:

- The authors state that a small amount of deception is involved, but don't explain why this is necessary. I am not against using deception, but it would be good to explain this.
- One potential issue in the task is that the authors may end up running into a floor effect for the fair condition. In particular for comparing EEG signals, this might be an issue.
- The information content part of the music stimuli is unclear and confusing. How is the information content determined and what are the implications of this in addition to whether the final note is expected or unexpected? Do unexpected notes have different information contents? If so, this would need to be accounted for in your statistical model.
- On page 7, line 4(7) the authors talk about 4 unfair offers, though, if I understand the task and materials correctly, these are not "offers" but are different amounts that player A chooses to steal/take from player B.
- From reading the materials, it is ambiguous whether the authors intend to look at choice behaviour as a binary outcome (take / leave) or only look at the choice behaviour as a continuous outcome from 0-100.
- Procedure and analysis-pipeline seem clear and sound. For the behavioural analysis, one may consider using a linear mixed effect model instead. Mixed effects model does not require to aggregate behaviour to a mean per subject, losing valuable information of individuals' behaviour. Mixed effects models generally are more powerful as you include the inter-individual variance and intra-individual variance.

- Are all trials the same length in duration? To avoid participants to finish the task quicker by choosing “leave”.
- Some suggestions for the instructions in Appendix A:
 - o The instructions of the role of the players in the game are not very clearly introduced before explaining the actions the player can take. Specifically, line 25 where it explains the actions of player A without any context of the role of player A and the game. I would suggest to include some sort of cover story as to why player A can steal from B. For example, by introducing that player A has given the possibility or power to take some of the divided chips from player B to increase player A's total amount of chips (and money). At the same time, player B cannot do anything. Then introduce player C's role.
 - o The introduction of the music part comes a bit unexpected and is quite obvious that it probably should have an effect on their behaviour. Perhaps the authors could try to better fit this in the game with a cover story so it doesn't become so obvious.
 - o The Take action becomes a bit confusing with the “How much do you want to spend” screen, as in this part it is not clear how much 30 chips spend means. I would suggest to better integrate the take action, spending of own chips and how that affects punishment (x3) in the text before showing an example of the trial.

Botvinick, M. M., Braver, T. S., Barch, D. M., Carter, C. S., and Cohen, J. D. (2001). Conflict monitoring and cognitive control. *Psychol. Rev.* 108, 624–652. doi: 10.1037//0033-295X.108.3.

Author's Response to Decision Letter for (RSOS-190048.R0)

See Appendix B.

RSOS-190048.R1 (Revision)

Review form: Reviewer 1

Is the language acceptable?

Yes

Do you have any ethical concerns with this paper?

No

Have you any concerns about statistical analyses in this paper?

No

Recommendation?

Accept in principle

Comments to the Author(s)

Dear Elisa and Claudia,

my compliments for the revision. I still think the music violations are not violating much. They actually sound more interesting than the rest :-)

Perhaps in naive listeners they may sound more violating.

Good luck with the data collection.

Review form: Reviewer 2

Is the language acceptable?

Yes

Do you have any ethical concerns with this paper?

No

Have you any concerns about statistical analyses in this paper?

No

Recommendation?

Accept in principle

Comments to the Author(s)

I believe the authors have addressed all my concerns and comments in an adequate way.

I have one minor comment on the mixed effect model in the r-script:

Once the authors have collected the data, I would suggest that when running mixed effects models, they first test whether their mixed effect models should at least include by-participant random slopes for the within-subject predictors (Fairness expectation, Music expectancy). They can do this by performing model comparison (with `anova()` function), to justify what their maximal model is given their data. At the moment in their r-script, the models are all random-intercept only models. The authors can do the model comparison for all combinations of random slopes. The same holds for the random intercept of Block, if the authors expect random variation in slopes for Division and InfCont by block.

References:

1. Matuschek H, Kliegl R, Vasishth S, Baayen H, Bates D (2017) Balancing Type I error and power in linear mixed models. *J Mem Lang*. doi:10.1016/j.jml.2017.01.001.
2. Bates D, Kliegl R, Vasishth S, and Baayen, H. (2015). "Parsimonious Mixed Models." ArXiv:1506.04967. <http://arxiv.org/abs/1506.04967>.
3. Barr DJ, Levy R, Scheepers C, Tily HJ (2013) Random effects structure for confirmatory hypothesis testing: Keep it maximal. *J Mem Lang* 68(3):255–278.

I would like to wish the authors good luck with their study.

Decision letter (RSOS-190048.R1)

08-May-2019

Dear Dr Carrus,

On behalf of the Editors, I am pleased to inform you that your Stage 1 Registered Report RSOS-190048.R1 entitled "Does unfairness sound wrong? A cross-domain investigation of expectations in music and social decision-making." has been accepted in principle for publication in Royal

Society Open Science subject to minor revision in accordance with the referee and editor suggestions. Please find their comments at the end of this email.

The reviewers and handling editors have recommended publication, but also suggest some minor revisions to your manuscript. Therefore, I invite you to respond to the comments and revise your manuscript.

Please you submit the revised version of your manuscript within 7 days (i.e. by the 16-May-2019). If you do not think you will be able to meet this date please let me know immediately.

When submitting your revised manuscript, you will be able to respond to the comments made by the referees and you should upload a file "Response to Referees". You can use this to document any changes you make to the original manuscript. In order to expedite the processing of the revised manuscript, please be as specific as possible in your response to the referees.

Full author guidelines can be found here <https://royalsocietypublishing.org/rsos/registered-reports>.

on behalf of Prof Chris Chambers (Subject Editor, Royal Society Open Science)
openscience@royalsociety.org

Editor Comments to Author (Professor Chris Chambers):

The two original reviewers responded positively to the revision. There is one remaining statistical issue to consider from Reviewer 2. Once this is addressed in a response, Stage 1 acceptance should be forthcoming without requiring further in-depth review.

Reviewer comments to Author:
Reviewer: 1

Comments to the Author(s)
Dear Elisa and Claudia,

my compliments for the revision. I still think the music violations are not violating much. They actually sound more interesting than the rest :-)
Perhaps in naive listeners they may sound more violating.

Good luck with the data collection.

Reviewer: 2

Comments to the Author(s)

I believe the authors have addressed all my concerns and comments in an adequate way.

I have one minor comment on the mixed effect model in the r-script:

Once the authors have collected the data, I would suggest that when running mixed effects models, they first test whether their mixed effect models should at least include by-participant random slopes for the within-subject predictors (Fairness expectation, Music expectancy). They can do this by performing model comparison (with `anova()` function), to justify what their maximal model is given their data. At the moment in their r-script, the models are all random-intercept only models. The authors can do the model comparison for all combinations of random slopes. The same holds for the random intercept of Block, if the authors expect random variation in slopes for Division and InfCont by block.

References:

1. Matuschek H, Kliegl R, Vasishth S, Baayen H, Bates D (2017) Balancing Type I error and power in linear mixed models. *J Mem Lang*. doi:10.1016/j.jml.2017.01.001.
2. Bates D, Kliegl R, Vasishth S, and Baayen, H. (2015). "Parsimonious Mixed Models." ArXiv:1506.04967. <http://arxiv.org/abs/1506.04967>.
3. Barr DJ, Levy R, Scheepers C, Tily HJ (2013) Random effects structure for confirmatory hypothesis testing: Keep it maximal. *J Mem Lang* 68(3):255-278.

I would like to wish the authors good luck with their study.

Author's Response to Decision Letter for (RSOS-190048.R1)

See Appendix C.

Decision letter (RSOS-190048.R2)

28-May-2019

Dear Dr Carrus

On behalf of the Editor, I am pleased to inform you that your Manuscript RSOS-190048.R2 entitled "Does unfairness sound wrong? A cross-domain investigation of expectations in music and social decision-making," has been accepted in principle for publication in *Royal Society Open Science*.

You may now progress to Stage 2 and complete the study as approved. Before commencing data collection we ask that you:

- 1) Update the journal office as to the anticipated completion date of your study.
- 2) Register your approved protocol on the Open Science Framework (e.g. using the dedicated RR registration interface at <https://osf.io/rr>) or other recognised repository, either publicly or privately under embargo until submission of the Stage 2 manuscript. Please note that a time-

stamped, independent registration of the protocol is mandatory under journal policy, and manuscripts that do not conform to this requirement cannot be considered at Stage 2. The protocol should be registered unchanged from its current approved state, with the time-stamp preceding implementation of the approved study design.

Following completion of your study, we invite you to resubmit your paper for peer review as a Stage 2 Registered Report. Please note that your manuscript can still be rejected for publication at Stage 2 if the Editors consider any of the following conditions to be met:

- The results were unable to test the authors' proposed hypotheses by failing to meet the approved outcome-neutral criteria.
- The authors altered the Introduction, rationale, or hypotheses, as approved in the Stage 1 submission.
- The authors failed to adhere closely to the registered experimental procedures. Please note that any deviations from the approved experimental procedures must be communicated to the editor immediately for approval, and prior to the completion of data collection. Failure to do so can result in revocation of in-principle acceptance and rejection at Stage 2 (see complete guidelines for further information).
- Any post-hoc (unregistered) analyses were either unjustified, insufficiently caveated, or overly dominant in shaping the authors' conclusions.
- The authors' conclusions were not justified given the data obtained.

We encourage you to read the complete guidelines for authors concerning Stage 2 submissions at <http://rsos.royalsocietypublishing.org/content/registered-reports>. Please especially note the requirements for data sharing, reporting the URL of the independently registered protocol, and that withdrawing your manuscript will result in publication of a Withdrawn Registration.

Please note that Royal Society Open Science will introduce article processing charges for all new submissions received from 1 January 2018. Registered Reports submitted and accepted after this date will ONLY be subject to a charge if they subsequently progress to and are accepted as Stage 2 Registered Reports. If your manuscript is submitted and accepted for publication after 1 January 2018 (i.e. as a full Stage 2 Registered Report), you will be asked to pay the article processing charge, unless you request a waiver and this is approved by Royal Society Publishing. You can find out more about the charges at <http://rsos.royalsocietypublishing.org/page/charges>. Should you have any queries, please contact openscience@royalsociety.org.

Once again, thank you for submitting your manuscript to Royal Society Open Science and we look forward to receiving your Stage 2 submission. If you have any questions at all, please do not hesitate to get in touch. We look forward to hearing from you shortly with the anticipated submission date for your stage two manuscript.

on behalf of Professor Chris Chambers (Registered Reports Editor, Royal Society Open Science)
openscience@royalsociety.org

Author's Response to Decision Letter for (RSOS-190048.R2)

See Appendix D.

RSOS-190048.R3 (Revision)

Review form: Reviewer 1

Is the manuscript scientifically sound in its present form?

Yes

Are the interpretations and conclusions justified by the results?

No

Is the language acceptable?

Yes

Do you have any ethical concerns with this paper?

No

Have you any concerns about statistical analyses in this paper?

Yes

Recommendation?

Accept with minor revision

Comments to the Author(s)

I read the paper by Claudia Civai et al. and I think authors did a great job. Congratulations. It is not easy to collect so many data with a EEG study!

That said, I have still a few comments. Overall, it looks to me authors are still very much in love with the original hypothesis. Although results do not show the hypothesized effect, authors discuss "marginally significant" results and make speculations about possible trends. These speculations, in my opinion, should be dropped all together. Here we have a solid study that had the statistical power (ie the fuel) to detect effects of certain sizes (ie to travel certain distances). There is simply not enough fuel to travel the various possible effects the authors see in the results. One additional point. I'm not an "effect size" expert, but If you look at the formula of the partial eta square, it ranges from 0 to 1. Effects sizes in the range of 0.05 looks to me very small...

A few specific comments:

- Abstract. Please write the abstract in such a way that is crystal clear to the reader the distinction between hypothesis driven analysis/results and exploratory results. There should be no confusions when reading the abstract.
- Results section. Drop "marginally significant" for $p=.06/.07$ etc. There is not such a thing like a marginally significant result. Drop also words like "trend" or "tendency".
- Figures. Please convert barplots into boxplots with mean on top

ps: Sorry for the late review but I received the ms when I was on holiday.

Decision letter (RSOS-190048.R3)

Dear Dr Carrus:

On behalf of the Editor, I am pleased to inform you that your Stage 2 Registered Report RSOS-190048.R3 entitled "Does unfairness sound wrong? A cross-domain investigation of expectations in music and social decision-making." has been deemed suitable for publication in Royal Society Open Science subject to minor revision in accordance with the referee suggestions. Please find the referees' comments at the end of this email.

The reviewers and Subject Editor have recommended publication, but also suggest some minor revisions to your manuscript. Therefore, I invite you to respond to the comments and revise your manuscript.

Please also ensure that all the below editorial sections are included where appropriate -- if any section is not applicable to your manuscript, please can we ask you to nevertheless include the heading, but explicitly state that the heading is inapplicable. An example of these sections is attached with this email.

- Ethics statement

- Data accessibility

If you wish to submit your supporting data or code to Dryad (<http://datadryad.org/>), or modify your current submission to dryad, please use the following link:
[http://datadryad.org/submit?journalID=RSOS&manu=\(Document not available\)](http://datadryad.org/submit?journalID=RSOS&manu=(Document not available))

- Competing interests

- Authors' contributions

AB carried out the molecular lab work, participated in data analysis, carried out sequence alignments, participated in the design of the study and drafted the manuscript; CD carried out

the statistical analyses; EF collected field data; GH conceived of the study, designed the study, coordinated the study and helped draft the manuscript. All authors gave final approval for publication.

- Acknowledgements

- Funding statement

Because the schedule for publication is very tight, it is a condition of publication that you submit the revised version of your manuscript within 7 days (i.e. by the 27-Aug-2020). If you do not think you will be able to meet this date please let me know immediately.

Please note that Royal Society Open Science will introduce article processing charges for all new submissions received from 1 January 2018. Registered Reports submitted and accepted after this date will ONLY be subject to a charge if they subsequently progress to and are accepted as Stage 2 Registered Reports. If your manuscript is submitted and accepted for publication after 1 January 2018 (i.e. as a full Stage 2 Registered Report), you will be asked to pay the article

processing charge, unless you request a waiver and this is approved by Royal Society Publishing. You can find out more about the charges at <https://royalsocietypublishing.org/rsos/charges>. Should you have any queries, please contact openscience@royalsociety.org.

on behalf of Professor Chris Chambers (Associate Editor) and Chris Chambers
(Registered Reports Editor, Royal Society Open Science)
openscience@royalsociety.org

Associate Editor Comments to Author (Professor Chris Chambers):

Associate Editor: 1

Comments to the Author:

One of the previous Stage 1 reviewers was available to assess the Stage 2 manuscript and I have also read it myself. I agree with the reviewer's positive assessment and also the critical points -- primarily that the conclusions must be dominantly shaped by the outcomes of the preregistered hypothesis tests, with the distinction between preplanned and exploratory outcomes made clear at all times (particularly in the Abstract). In all other respects the submission is a fine exemplar of the RR format. Following minor but careful revision to address this point throughout the manuscript, full acceptance is likely to be forthcoming without requiring further in-depth review.

Associate Editor: 2

Comments to the Author:

(There are no comments.)

Comments to Author:

Reviewer: 1

Comments to the Author(s)

I read the paper by Claudia Civai et al. and I think authors did a great job. Congratulations. It is not easy to collect so many data with a EEG study!

That said. I have still a few comments. Overall, it looks to me authors are still very much in love with the original hypothesis. Although results do not show the hypothesized effect, authors discuss "marginally significant" results and make speculations about possible trends. These speculations, in my opinion, should be dropped all together. Here we have a solid study that had the statistical power (ie the fuel) to detect effects of certain sizes (ie to travel certain distances). There is simply not enough fuel to travel the various possible effects the authors see in the results. One additional point. I'm not an "effect size" expert, but If you look at the formula of the partial eta square, it ranges from 0 to 1. Effects sizes in the range of 0.05 looks to me very small...

A few specific comments:

- Abstract. Please write the abstract in such a way that is crystal clear to the reader the distinction between hypothesis driven analysis/results and exploratory results. There should be no confusions when reading the abstract.
- Results section. Drop "marginally significant" for $p=.06/.07$ etc. There is not such a thing like a marginally significant result. Drop also words like "trend" or "tendency".
- Figures. Please convert barplots into boxplots with mean on top

ps: Sorry for the late review but I received the ms when I was on holiday.

Author's Response to Decision Letter for (RSOS-190048.R3)

See Appendix E.

Decision letter (RSOS-190048.R4)

Dear Dr Carrus,

It is a pleasure to accept your manuscript entitled "Does unfairness sound wrong? A cross-domain investigation of expectations in music and social decision-making." in its current form for publication in Royal Society Open Science.

Appendix A

London, 9th January, 2019

Dear Professor Chris Chambers,

This letter accompanies the resubmission of our registered report titled “Does unfairness sound wrong? A cross-domain investigation of expectations in music and social decision-making” with ID RSOS-182045.

We have now considered all the comments made by the Editor, made the required changes and addressed the issues raised. We are including the responses to the Editor’s comments below.

We hope that you find this new and improved submission of our Registered Report suitable for your journal.

Sincerely,

Dr Elisa Carrus and Dr Claudia Civai

School of Applied Science | Division of Psychology, London South Bank University, 103 Borough Road | London SE1 0AA

Response to Decision Letter for Registered Report pre- Stage 1

Editor's Comments to the Author:

Should you decide to resubmit, please address the following issues:

1. The Introduction feels very long. Please consider whether all of the content prior to the statement of aims is necessary. If it is, I would recommend shortening paragraphs to aid readability and anticipate the aims much sooner (e.g. at the end of the first paragraph). Please also supply an abstract summarising the research question, rationale, aims and methods.

We thank the Editor for this comment. We have now re-organised and shortened the introduction as follows:

- The aims have been made explicit at the end of the first paragraph (p. 2, line 32), as suggested;
- The detailed description of the social decision-making task and the rationale for using this task have been moved from the introduction to the Methods section (pp. 5-6 from line 22)
- The hypothesis has been simplified, as we are now focusing on the interaction effect, which is our effect of interest (p. 4, line 8-28);
- The abstract has been added at the beginning of the manuscript

We believe that this section has improved and it is now clearer and more readable than the previous manuscript.

2. A clearer mapping is required between the hypotheses, power analyses, statistical test, and theoretical implications of the potential outcomes. Suggest listing presenting hypotheses in a more explicit list format at the end of the introduction and then making clear in the Analysis section (again, using a list) which hypothesis is associated with which statistical test(s), and crucially, which outcomes will confirm or disconfirm the hypothesis. You may also wish to summarise this in tabular format across all hypotheses to aid information synthesis. Every preregistered hypothesis must have its own or sampling plan (e.g. power analysis calculated analytically or through simulations) linked to the specific test. Many of your tests are not directly linked to a hypothesis or power analysis (e.g. the LME analysis). Where multiple analyses are used to assess one hypothesis, it must be crystal clear which outcomes (or combination of outcomes) would confirm or disconfirm the hypothesis in question.

We thank the Editor for the useful suggestions. We believe that two important points were raised in this comment:

- 1) *The need for a clearer mapping of hypothesis, statistical test, and outcomes:*
We have now restricted our hypothesis to our effects of interest, which are the two interactions between music and social decision-making, both in the behavioural and in the EEG data. The main effects have been listed as positive controls (p. 9) rather than as main hypotheses. The two analyses (ANCOVAs) and the potential outcomes have been listed and mapped onto the

hypotheses (p. 9, lines 2-23). Since the ANCOVA is the most straightforward analysis that can be performed to address our hypothesis, we have removed the other analyses from the plan.

2) *The need for a sampling plan:*

We agree with the comment above and we have therefore addressed this issue in the way we think is most appropriate for this context (p. 5, lines 1-9). We have now reduced our hypotheses to target the interaction effect, and we have initially referred to the literature to find the effect sizes reported for a similar interaction observed in a similar paradigm, conditions and populations and using similar statistical tests. For the effect of interest (the interaction music x social decision-making), the smallest effect size reported is $\eta_p^2 = .20$. A power analysis run using this effect size shows that 34 participants will be needed to reach power of .80. It is worth noting that G*Power offers a conservative sample size estimate because it assumes independence of treatments in the data, and thus we are confident that our chosen sample size of 34 participants is adequate to address the experimental question.

We understand that referring to the literature offers a biased estimated guess for our sampling plan; however, we believe it to be a more suitable choice compared to using standardised effect sizes, which would only offer a random guess of what we think we will find. We would like to specify that, in the context of the present study, we are not interested in reaching a specific effect size but rather in testing whether a difference in our conditions exists (Keppel, 1973).

3. Related to (2), contingencies for additional analyses are in many places too vague, e.g. "If our sample shows enough variability in initial expectations (as measured at the beginning of the experiment), participant's initially recorded expectations on the offender's behaviour will be considered as a covariate (2x2 repeated measures ANCOVA)." What will count as sufficient variability?

We agree with the Editor that the contingencies for the analysis proposed were too vague. We have now simplified our analysis plan, and, since we expect a variability in the participants' initial expectations of the offender's behaviour, we have opted to run an ANCOVA, considering this initial expectation as a covariate. If no variability is shown, i.e., all participants report the same level of expectation, then the addition of the covariate will lose any meaning and will be removed.

4. For added clarity, you might consider presenting the hypotheses in graphic (visual) form to make the directionality of the predicted effects clear in every case.

Since our hypotheses have been simplified, we hope that the predicted effects are now clearly conveyed.

5. It is unclear if the sample size of 30 will be sufficient to provide a sensitive test of all hypotheses. You should justify the sampling plan according to the smallest effect size of theoretical interest, rather than using single point estimates of observed effect sizes from previous studies (which are likely biased in a positive direction). At present, reviewers are likely to consider N=30 to be insufficiently justified and thus too small given the large number of statistical tests and predictions.

We believe this comment refers to the issue raised previously, so we refer the Editor/reviewers to our responses for point 2 (sampling plan). As mentioned before, we completely agree with the Editor that our sampling plan had not been properly justified and explicitly supported by precise calculations. Since we have simplified our hypothesis and we now focus on the interaction effect, we have performed a power analysis to determine the sample size based on the smallest and most similar interaction effect of interest found in the literature, i.e., $\eta_p^2 = .20$ (Carrus et al., 2013). The result of this analysis was 34 participants, which is a conservative estimate because G*Power assumes independence of treatments in the data. Therefore, as previously specified, we are confident that our chosen sample size of 34 participants is adequate to address the experimental question.

7. One of the key criteria that reviewers are asked to assess at Stage 1 is "Whether the authors have considered sufficient outcome-neutral conditions (e.g. absence of floor or ceiling effects; positive controls) for ensuring that the results obtained are able to test the stated hypotheses", and successfully passing such tests is an editorial criterion at Stage 2 following completion of the study (see <http://rsos.royalsocietypublishing.org/registered-reports>). Please make clear which tests will serve, e.g. as positive controls for the effect of fairness and music expectedness, as well as any other outcome neutral tests to e.g. confirming quality of EEG data.

The Editor raised another important point and we apologise for the oversight. The outcome-neutral conditions and positive controls have now been listed for both behavioural and EEG data (p. 9 lines 25-35) as below:

"Before testing our hypothesis, we will run analysis to test outcome-neutral conditions.

To confirm the quality of the EEG data, we expect:

1. An N1 component locked to the onset of the final note;
2. an MFN component locked to the onset of the division

To ensure that the data are suitable to test our hypothesis, we expect:

1. The rate of punishment for unfair divisions to be significantly higher than rate of punishment for fair divisions;
2. a larger N1 for unexpected vs. expected notes;
3. a difference in the MFN between unfair vs. fair divisions, with a larger MFN for unfair divisions."

8. Exclusion criteria for data within participants, and for participants within the sample: please ensure that these are comprehensively pre-specified as it is generally not possible to adjust these for pre-registered analyses after provisional acceptance has been awarded. Please also make clear whether any excluded participants would be replaced.

We have now included this information in the manuscript (p.8 from line 6) as below:

“Data cleaning and data removal

All behavioural data will exclude any trials with a reaction time that is larger or smaller than 2 standard deviations. Any null responses will be removed.

EEG data will be pre-processed as indicated below. EEG data associated with the removed behavioural data above will also be removed.

The data for any participant who does not complete the study will not be used. Any rejected participant will be replaced in accordance with the sampling plan.”

9. Method of eyeblink removal in EEG data must be objectively reproducible. Subjective method (visual inspection) is insufficient unless performed by independent, blinded analysts using a clearly described reproducible methodology for which you can demonstrate high inter-rater reliability. Most RRs proposing the use of EEG use an objective algorithm to exclude eyeblinks to ensure reproducibility.

We have now included this information in the manuscript (p.8 from line 11).

“EEG data pre-processing

Prior to data analysis, the data will be pre-processed using the standardised PREP pipeline³⁹. The data epochs representing single experimental trials will be extracted around the onset of the division (-500 msec to 1000 msec, with $t=0$ as the onset of the last note/word). Correction of eye-blink artefacts will be carried out using the automated algorithm ADJUST implemented using the SASICA plugin in EEGLAB⁴⁰. The output of the rejection will be checked against the recommendations provided by [41] to increase reproducibility and objectivity.”

10. To ensure a clear separation between pre-registered confirmatory analyses and exploratory post hoc analyses, any mention of exploratory analysis should be removed from the Stage 1 manuscript and introduced at Stage 2 (after data collection) in the "Exploratory Analyses" section of the Results. Alternatively, if these exploratory analyses address specific hypotheses then they can be kept in the Stage 1 submission but must be fully elaborated and associated with a sampling plan (e.g. power analysis).

The proposed exploratory analyses have now been removed from the manuscript.

References

Carrus, E., Pearce, M. T., & Bhattacharya, J. (2013). Melodic pitch expectation interacts with neural responses to syntactic but not semantic violations. *Cortex*, 49(8), 2186-2200.

Keppel, G. (1973). *Design and analysis: a researcher's handbook*. Prentice-Hall: Englewood Cliffs, NJ.

Appendix B

Dear Dr Carrus,

The Editors assigned to your Stage 1 Registered Report ("Does unfairness sound wrong? A cross-domain investigation of expectations in music and social decision-making.") have now received comments from reviewers. We would like you to revise your paper in accordance with the referee and editors suggestions which can be found below (not including confidential reports to the Editor). Please note this decision does not guarantee eventual acceptance.

on behalf of Chris Chambers (Registered Reports Editor, Royal Society Open Science)
openscience@royalsociety.org

Associate Editor Comments to Author (Professor Chris Chambers):

Two expert reviewers have now assessed the manuscript. Both see merit in the proposal, but also raise concerns that cut across the full range of Stage 1 review criteria, from the rationale of the proposed hypotheses, to the clarity and level of detail in the methods (including the power analysis), to the justification of specific methodological decisions and the consideration of necessary controls. Please also ensure that any digital materials are available to the reviewers. Such materials will be required to be publicly archived in the event of final Stage 2 submission, but as per the comments would clearly be useful to the reviewers earlier. Given these assessments, a Major Revision is recommended.

Comments to Author:

We thank the Editor and both Reviewers for taking the time to review our registered report. We believe the changes suggested have significantly improved our report and

study, and we hope our comments have satisfactorily addressed the issues raised. Below we provide a comment-by-comment response to each of the reviewer.

Reviewer: 1
Comments to the Author(s)

I read the stage 1 by Carrus and Civai. This is the first time I review a registered report. Thank you for the opportunity and apologies in advance if I make some mistake. My recommendation is "major revision" because I ask authors a lot of work (in particular R and matlab scripting).

The authors want to investigate with a bio measure and a behavioural measure whether a violation in one field (social) interacts with a violation in another field (music).

Hypothesis.

The hypothesis is driven by that part of the current literature that suggests a relationship between music and language. I do not share much that view, but, definitely, things are out there so I have nothing to complain about the hypothesis. Perhaps, the only critic here is about how strong can be this relationship. Therefore, how strong we expect to be the relationship between the violation in music and the social violation (see below). My best guess is that this relationship is weak and small.

We thank the reviewer for this comment. Previous data collected by Carrus et al. show that the strength of the relationship between music and language is fairly high (partial eta squared (η_p^2) = 0.24); nevertheless, we recognise that the relationship between social domain (high-order cognition) and music (low-order cognition) is potentially weaker, as also pointed out by Reviewer#2. We did take this hypothesis into account when calculating sample size, but, following Reviewer#1's advice, we have now been even more conservative in estimating our effect.

Method.

I list here a set of critics and (when it is possible) suggestions.

*The first problem seems to me the number of participants. Given that I suspect the relationship to be small, I was surprised to see that the power calculation returned 34 subjects. I even downloaded G*power and tried to calculate N myself. Unfortunately, I found parameters that I did not know how to fill in. If I fill all the info reported in the paper, the software tells me "groups must be more than 2" and, in addition, there is another parameter I can't fill in because it is not reported by the authors. I'm sure the problem it's me. However, I would like authors to explain me step by step how N was calculated. I would suggest to write an R script (I will suggest this later also) but, in case, even a video or a screenshot that show the N is calculated should be fine.*

I suggest also other strategies to calculate N. There is a large debate now about power calculation. Many suggest that power calculation has no sense given that data in literature may not be that reliable. Therefore, any power calculation is unreliable. A conservative approach could be that of targeting the N that enables to reveal the smallest possible effect of interest for the given effect. I explain myself. Let suppose authors think

the smallest effect they would like to detect is $d=.30$ (i.e., a small effect). They should calculate N with this effect size in mind. In this way, the experiment would reveal the effect even if the effect is small. This is a very conservative approach.

In any case, 34 subjects to test the relationship between a social violation and a music violation (two very far fields) seems very small.

We thank the reviewer for the opportunity to clarify the way in which we calculated our sample size. From our understanding, the optimal way to estimate sample size for repeated-measure within-subject ANOVA with multiple factors is to run a simulation on pilot data. Unfortunately, we do not have pilot data, and, as the reviewer points out, the literature might not be reliable. Moreover, an analysis of the documentation and statistical blogs dedicated to power calculation with G*Power (<https://stats.stackexchange.com/questions/59235/repeated-measures-within-factors-settings-for-gpower-power-calculation>) showed that the programme does not fully support repeated-measure within-subject ANOVA with multiple factors. Following suggestions provided by these sources, we used the option of ANCOVA fixed effects (because we also have a covariate in our design): given that this analysis assumes complete independence between the measures, it is more conservative when estimating sample size from effect size. The estimated effect size of the interaction ($\eta_p^2 = 0.2$) was obtained from a dataset of a previous study conducted by Carrus et al (in preparation); we acknowledge that this value indicates the effect size of the interaction between music and language, but this is the theoretically closest effect that we could refer to. From this analysis, we obtained 34 participants (Figure 1).

Parameter specification:

- Effect size $F = 0.5$ (corresponding to $\eta_p^2 = 0.2$; using the “calculate” button on the left, a drawer will appear where η_p^2 can be directly inputted and f calculated and transferred to the main window)
- $\alpha = 0.05$
- Power = 0.80
- Numerator $df = 1$ (calculated as $(2-1)*(2-1)$, given that our measures have 2 levels each)
- Number of groups = 4 (pretending that our design is between-participants)
- Number of covariables = 1

Figure 1. G*Power calculation of sample size using ANCOVA fixed-effects and considering $\eta_p^2 = 0.2$.

Other suggestions proposed by contributors of Stackexchange included using repeated measures, within-subjects ANOVA, with “4” as number of measurements for a 2x2 ANOVA (i.e., the number of conditions of our within-subject design). Note that, in “Options” (bottom left button), you must select “Effect size specification: as in SPSS”. This communicates to G*Power that the non-sphericity correction error is taken into account in the effect-size measure (as suggested by Soderberg et al, 2017; <https://osf.io/kmbg6/>). Using this analysis, a $\eta_p^2 = 0.2$ returned $N = 17$. Considering a medium effect size ($\eta_p^2 = .06$, corresponding to $f(u) = .25$), we obtain $N = 60$ (Figure 2); considering a small effect ($\eta_p^2 = .01$), $N = 363$.

Parameter specification:

- Effect size $f(U) = 0.25$ (corresponding to $\eta_p^2 = 0.06$)
- $\alpha = 0.05$
- Power = 0.80
- Number of groups = 1
- Number of measurements = 4 (pretending that our design is between-participants)
- Nonsphericity correction $\epsilon = 1$ (automatically calculated by choosing the “Options” button, and choosing “as in SPSS”)

Figure 2. G*Power calculation of sample size using ANOVA repeated measures, within factors, and considering medium $\eta_p^2 = 0.06$.

Considering the relatively exploratory nature of the study, the resources available for this study, and the fact that now we have adopted a more sensitive analysis (mixed-model), if the reviewer accepts, we would like to propose to opt for the results returned by using the less conservative option “repeated measures, within-subjects ANOVA” with a medium size effect, and use a sample size of 60. We think that this could be a good compromise between the conservative approach of using the small effect size and the possibly too optimistic prediction that the size of the interaction between social norms and music would be equivalent to music x language interaction (even though the current sample size had already been calculated conservatively using a between-subject design).

Materials.

Another problem with the method is that materials are not available. I was curious (for example) to see the task and to listen to the melodies. I found no link to materials. Materials are just briefly described (By the way, I synthesized the score that is reported in the appendix and played the melody. Well, perhaps I have very a broad music taste. But the violation definitely did not violate me!). In synthesis, the experiment should be available so that we can try the task. Note that there is no description about the music stimulus. Which waveform? Sample rate? Bits? Listening level? Freefied or headphones? (it looks headphones from the picture).

We have now uploaded all material, including the melodies (.wav), which will have information about sample rate and bits, the E-prime task, and stimuli information as entered into the stimulus presentation software. The melodies will be presented via

earphones at a constant volume for all participants, as done in Carrus et al. (2013). We have also included a video of the task in case the reviewers do not have access to E-prime. We have added additional information about the melodies and the computational model used to derive the melodies in the manuscript (page 7, line 9-21 “Music Stimuli”). The parameters for the model and the melodies are from the same pool as those used in Carrus et al. (2013).

Analysis. Behavioural measure.

It is not clear what is the criterion to remove outliers in the behavioural measure. The mean of what? The mean of the participant, of the condition? The standard deviation of what? The sd of the participant, of the condition? The best way to sort everything out is to add a sample dataset (fake, with simulated data) and an R script that, from raw data to statistical results shows what authors intend to do. If this script included the graph too, it would be brilliant. Then, when data collection is over, authors would have simply to run the script on the real data. Script and fake-data should be available like the digital materials in the next submission. I would also comment section by section the various parts of the script (e.g. from raw data to means, removing outliers...)

We have now provided these scripts together with an explanation of step by step analysis. In the text, we have specified in more detail the criterion to remove the outliers:

“Behavioural data from each participant will be cleaned: for each participant, any trial with a RT that is larger or smaller than 2 standard deviations from the participant’s average will be excluded, as well as any null responses. The scripts have been uploaded and are available to download and review.”

Analysis. EEG.

This is a problem. And I’m not an EEG expert. So I asked a colleague that works with EEG. He told me the pipeline is good and standard and I trust him as well as the authors. The problem, as usual, is when the experimenter’s hand comes into play. There are two problems: reproducibility of the results and reliability of the results. Reproducibility. The colleague suggested me you should write down all the signals that will be discharged at the various stages of the analysis. And report also the stage of the pipeline together with the reason why the signal was discharged. This would guarantee the reproducibility of the result you will collect. Reporting the exclusion criteria would be fundamental here. I see you are using a matlab toolbox. My extra suggestion would be that to write a batch script of the analysis you do. The batch script should start from raw data up to final analysis and, of course, it should include outliers removal.

Another option I think you could try is to have a third independent party (e.g., a colleague that is working with eeg) that will be consulted to do the outlier removal. This colleague needs to be, of course, hypothesis-free. Like for the behavioural experiment, I would add an R script that from raw data takes analysis to final results, maybe a graph.

We understand the reviewers' concerns, especially given some of the potentially subjective decisions that must be made at the pre-processing stage. We have now added the script for the analysis, which the reviewer can see. In brief, in order to minimise subjective judgments and increase transparency, reliability and reproducibility, we will be implementing a standardised pipeline across all participants with minimal subjective intrusion from the researcher. Any signal discarded will also be reported and justified. The early pre-processing steps will be following the PREP pipeline (Bigdely-Shamlo, Mullen, Kothe, Su, & Robbins, 2015). Following this, ICA will be performed to correct for artifactual signal. To minimise subjective judgments, the removal of independent components will be carried out using MARA which allows for automatization of some of the steps. MARA will identify artifactual components to reject based on key features extracted using a machine learning algorithm. Although MARA allows for blind automatic removal of independent components, the artifactual components will be inspected and removed according to the exclusion criteria provided by the three features (see information and tutorial here: <https://irene.github.io/artifacts/>) as well as following these guidelines <https://labeling.ucsd.edu/tutorial/overview>. This information will be recorded for each participant for transparency and reproducibility. Following this step, the data will be visualised to ensure that no obvious artefactual signal remains. If any obvious artefacts remain, the ICs outputted by the ICA and MARA's recommendations will be reviewed again. Importantly, the above pipeline will be used across participants to ensure standardisation. At each step of the analysis, the signal discarded following ICA will be transparently reported and justified based on the recommendations above. Datasets processed in EEGlab and Fieldtrip will contain a history log of the steps taken.

The MATLAB script for the analysis has been uploaded with the submission.

Additional control measures.

I would control for the music aptitude of the participants. This could explain why the effect works or not. First of all, I would check whether there are amusic. To check for aptitude there are two options. First use an independent test (e.g., PROMS). But it takes 20 minutes. https://www.uibk.ac.at/psychologie/fachbereiche/pdd/personality_assessment/proms/ta-ke-the-test/mini-proms/index.html.en

In alternative use a questionnaire such as the goldsmith. <https://www.gold.ac.uk/music-mind-brain/gold-msi/>

Again on music samples. I understand violations will be theoretical violations. Are we sure participants perceive them as violations? I mean: is there any subjective measure of the "violationness" of each melody?

This measure (the PROMS, the questionnaire or the subjective rating of violation) should be taken into account or controlled for by the statistical analysis. Perhaps an estimate of the violationness of each sample would be best. This could be easily collected with a pre-study.

We understand the reviewer's point here and we agree that a measure like the PROMS or the Gold-MSI would be ideal to measure participants' variability in music aptitude. However, it is also possible to minimise this by excluding anyone with musical training to enhance the homogeneity of our sample, as done in previous studies investigating music expectations (e.g. Perruchet and Poulin-Charronnat, 2013; Guo and Koelsch, 2016). Furthermore, the melodies used are derived from a probabilistic computational model of melodic expectation (IDyOM – e.g. Pearce et al., 2005, 2010), which allows to objectively quantify the level of expectation of each melody using information content, thus providing an estimate as required above by the reviewer. More information about this has been added in the manuscript (page 6, line 7-21, under "Music Stimuli"). Importantly, this model predicts the level of expectation subjectively perceived based on a quantifiable measure of information content. Each final note in each melody is associated with a value of information content, whereby a higher information content is associated with a more unexpected melody. Melodies of this type had also been independently rated by a sample of participants ($n = 10$) for a previous study (Carrus et al, currently in preparation). Participants rated the expectancy of the final note of each melody, and high-probability notes (low information content) were rated as more expected than low-probability notes (high information content), $t(9) = 3.21, p < .05$. We believe these measures allow us to objectively take into account the variability in the perception of melodic expectation as well as minimise the variability within our participant sample.

In synthesis,

1. I want to understand how power was calculated ($N=34$ seems rather small to me) and, in particular, in a very conservative approach what N should be like.

We hope we have provided a good explanation and a satisfactory (and more conservative) change to the proposal.

2. I would add to the submission all the possible digital materials: stimuli, experiment, analysis scripts so that, in principle, the reader could replicate the experiment here and now, including the statistical analysis and the signal analysis.

We have now uploaded all the materials on Dryad here:

3. I would look for a control measure that somehow check whether the music violation is perceived as a violation. If any, even a subjective evaluation of a different group of subjects would be interesting.

We hope we have provided a satisfactory explanation of the issue

Apologies for this late review (I'm usually faster).

We thank the reviewer for the thorough review.

References

Bigdely-Shamlo, N., Mullen, T., Kothe, C., Su, K. M., & Robbins, K. A. (2015). The PREP pipeline: standardized preprocessing for large-scale EEG analysis. *Frontiers in neuroinformatics*, 9, 16.

Carrus, E., Pearce, M. T., & Bhattacharya, J. (2013). Melodic pitch expectation interacts with neural responses to syntactic but not semantic violations. *Cortex*, 49(8), 2186-2200.

Guo, S., & Koelsch, S. (2016). Effects of veridical expectations on syntax processing in music: Event-related potential evidence. *Scientific reports*, 6, 19064.

Perruchet, P., & Poulin-Charronnat, B. (2013). Challenging prior evidence for a shared syntactic processor for language and music. *Psychonomic bulletin & review*, 20(2), 310-317.

Soderberg, C. K., Clyburne-Sherin, A., Spitzer, M., Sullivan, I., & Smith, J. F. (2017, October 20). Part 1: Choosing the correct options when inputting partial eta squared. Retrieved from osf.io/kmbg6

Reviewer: 2.

Comments to the Author(s)

In this report the authors describe a research proposal with the aim to investigate whether violations in expectations of music alter sensitivity to violations in fairness and influence subsequent social behaviour. The authors base their idea on literature from music and language showing how unexpected music tones affect language processing, both semantic and syntactic, and literature in social decision-making showing how violating social norm expectations affect social behaviour. The main question they seek to answer is whether expectations in music underlie a basic mechanism of expectancy that is shared across different domains such as social decision-making. To test this, they want to examine how expectations in music influence processing of fairness (with EEG) and social behaviour (decision behaviour), using an existing third-party punishment game. Their hypotheses are exploratory with each providing potential interpretations, namely 1) resource competition and 2) priming.

Main comments:

The cross-domain approach the authors attempt to investigate is interesting and novel.

We thank the reviewer for the interest expressed.

However, I have some concerns about the plausibility of the proposed interpretations of the hypotheses and aim of the study. First, the rationale to study music expectations with

social decision-making is not entirely clear from the proposal. The authors explain how social norms can be seen as rules and expectations which are similar to language acquisition, but they do not explain why they chose to specifically investigate the relation with social decision making or how this may be relevant for social decision-making. Second, social decision-making such as fairness is a complex behaviour that involves higher-order cognition, whereas language and music (as the authors mention) involve low-order cognitive processes. Moreover, language may not be completely unrelated to melodic tones. Claims about potential shared mechanisms between these processes and how music expectations influence such complex social behaviour needs to be taken with caution.

We agree with the reviewer that any claim on shared mechanisms should be taken with caution; nevertheless, if the findings will suggest an influence of one domain onto the other, a common mechanism could be at least hypothesised. Following the reviewer's suggestions, we have now restructured the introduction and added more context. We have provided a short introduction where we explain that the investigation into the role played by expectations in cognition has spanned across cognitive domains, from perception and action to decision-making (p. 2, lines 3-6):

"The role that expectation and prediction play across all areas of cognition has been widely investigated, from perception and action to decision making^{1,2,3}. One way of studying expectations is by investigating systems that are highly structured and therefore governed by a system of rules."

Then, we have explicitly said that music and social decision-making are two different domains, which are based on different cognitive structures; nevertheless, we can still recognise similarities, in that both domains are based on norms (p. 3, lines 8-19):

"In general terms, we can draw a parallel between expectations in music and the social domain in that they are both governed by rules that are derived from exposure to the environment and inform our expectations, ultimately guiding our behaviour. One notable difference, however, is that the expectations in music are based on a system of rules that governs the way units are meaningfully combined into structures as they unfold over time (e.g. notes into melodies)⁴. This is not the case for expectations derived from social norms. Having said so, both systems are similar in that knowledge about the rules and norms in the two domains allows for the formation of expectations about what is to follow given a certain preceding context. In music, the listener may expect a certain note following a given melodic context; in the same way, an observer may expect a specific behaviour given a certain social scenario (e.g. reciprocate trust or re-establish fair outcomes)."

We have also explained more in detail why we deem interesting to investigate the relationship between low-order (music) and high-order cognitive processes, in particular social decision-making (p. 3, lines 21-26):

“This investigation will shed light on the domain-generality of the expectancy mechanism, offering an insight into whether there is a shared mechanism of detecting deviations from expectancy both in lower-level cognitive domains (music), and in higher-level domains (social decision-making). Moreover, it will further our understanding on how low-order cognitive processes such as those involved in music perception may influence complex high-order processes such as social decision-making²².”

The authors provide two accounts to explain possible results. One alternative hypothesis would be that when expectations are incongruent, an increase in conflict-monitoring occurs and as result more attention to the target after an incongruent trial (Botvinick et al., 2001). Here, for example, an expected melodic tone and unfair behaviour of player A yields incongruency in expectations which may result in increased conflict signal, increased decision time and increased punishment. To distinguish between priming and conflict, the authors could also look at the reaction time of the decision.

We agree with the reviewer’s hypothesis that a potential effect of incongruency in expectations could be tested by analysing reaction times, and we thank them for the suggestion. We now have added RTs to the list of the dependent variables we are interested in investigating.

Furthermore, incongruent expectations may induce mood differences and influence fairness decisions. To control for mood as an explanation, one could include a mood assessment in the task (not every trial, but on some occasions) or either before and in the breaks. This can be included as a predictor of non-interest.

We agree with the reviewer that mood associated with incongruency may explain behaviour. Nevertheless, we believe that a reliable measure of potential mood manipulation would be difficult to obtain: asking participants to rate their mood after every trial would be far too disruptive for the experimental flow, but mood rating during the breaks could be influenced by a number of different variables, such as experimental distress or boredom. Hence, we do not deem this experimental set up to be particularly suitable for measuring the impact of mood or emotional reactivity on fairness decisions. However, we are currently planning a second EEG study aimed at investigating the influence of emotional reactivity on fairness decisions.

The design of the study allows to test the stated hypotheses.

Comments on description of the methods, experimental procedures and analysis pipeline:

- *The authors state that a small amount of deception is involved, but don’t explain why this is necessary. I am not against using deception, but it would be good to explain this.*

We have now specified in the main text why the deception is necessary. Participants are told that divisions are decided by real people who played beforehand, whereas they are, in

fact, a-priori established. This is necessary in order to make sure that each participant sees the same number of trials for each condition. Participants believe that their choice is going to actually influence the other players' payoffs, when, in fact, it is not, given that the other players are not real people; moreover, at the end, participants are paid a fixed amount, which is higher than what they would have expected in order to compensate for the deception.

- *One potential issue in the task is that the authors may end up running into a floor effect for the fair condition. In particular for comparing EEG signals, this might be an issue.*

We agree with the reviewer on this. The floor effect could be a problem for the analysis of the choice when comparing punishing choices in fair vs unfair divisions. Now we have modified our design and our analysis plan, and it should not be a problem anymore. First of all, following Reviewer#2's advice, we have added RTs as dependent variable, which is expected to be more sensitive in capturing behavioural variation. Secondly, we are now using a mixed-effect model (see below), de-facto removing the necessity to compare fair vs unfair divisions by using unfairness level of the division as a linear predictor of the choice.

- *The information content part of the music stimuli is unclear and confusing. How is the information content determined and what are the implications of this in addition to whether the final note is expected or unexpected? Do unexpected notes have different information contents? If so, this would need to be accounted for in your statistical model.*

Thank you for raising this issue. We have now added more information about this measure in the manuscript and hope that this is now clearer (page 6, line 7-21, "Music Stimuli"). In brief, the melodies are derived from a computational model of melodic expectation, which is a variable n-gram model that learns from a corpus of melodies. This means that the model represents the musical experience of a typical Western listener, based on their typical exposure to music. This model indeed predicts listeners' expectations, but what is useful about this model is that we can quantify expectation for each note, and this is measured in terms of information content, which tells us the degree to which a note appearing in a certain context is unexpected (McKay 2003). Each melody has five notes and the final note is associated with a value of information content. The smaller the value of information content, the more expected the final note. Therefore, unexpected notes have a higher value of information content than expected notes. More information about this model can be found from Pearce and Wiggins (2012) more recently, but also others (Pearce, 2005; Pearce et al, 2010). Because we have now changed our analysis to include linear mixed effects models (see below), the variability of information content within the melody set will be entered in our statistical model and the degree of melodic expectation taken into account.

- *On page 7, line 4(7) the authors talk about 4 unfair offers, though, if I understand the task and materials correctly, these are not "offers" but are different amounts that player A*

chooses to steal/take from player B.

We thank the Reviewer for noticing the mistake, which has now been corrected.

- *From reading the materials, it is ambiguous whether the authors intend to look at choice behaviour as a binary outcome (take / leave) or only look at the choice behaviour as a continuous outcome from 0-100.*

The original idea explained in the manuscript was to perform an ANCOVA with the percentage of punishing choices as dependent variable (i.e., continuous outcome 0-100). Nevertheless, we have now modified our analysis plan and a logistic mixed-effect model will be used to predict the binary outcome (see below).

- Procedure and analysis-pipeline seem clear and sound. For the behavioural analysis, one may consider using a linear mixed effect model instead. Mixed effects model does not require to aggregate behaviour to a mean per subject, losing valuable information of individuals' behaviour. Mixed effects models generally are more powerful as you include the inter-individual variance and intra-individual variance.

We completely agree with the reviewer on the benefit of using a mixed-effect model, which, as a matter of fact, was in our original plan. The reason why we opted for a repeated-measure within-participants ANCOVA analysis on the behavioural data was to mirror the ERP analysis, which involved averaging trials for each participant for each condition. However, in light of further considerations and the reviewer's suggestions, we have now decided to use a mixed-effect model for our main analysis.

We are now considering 2 predictors:

- Music expectancy, a continuous variable based on the information content of each melody (as explained above, and now also in the main text)
- Fairness expectancy, an interval variable based on how many chips player A is taking from player B (0, 25, 50, 75, 100)

And their predictive power on:

- Choice (binary outcome)
- Amount spent to punish (continuous)
- RTs (continuous)
- MFN amplitude (continuous)

The initial fairness expectation will also be added as a covariate in the model

Please note that this trial-by-trial analysis of the EEG signal will be conducted after having identified the MFN time-window using a more traditional ERP analysis, in which we compare the average signal, calculated for each participant for each condition, in the fair and extreme unfair (100) conditions (30 trials per condition).

The mixed-effect analysis will tell us the extent to which the levels of expectancy in music and fairness, and their interaction, predict our DVs.

Because of this change in the analysis plan, we have now diminished the number of trials. Since we do not have to compare fair to unfair division anymore, but we use fairness level as a predictor of the behavioural and neural outcomes, we have reduced the number of fair divisions to make it compatible with the number of divisions in the other fairness conditions: this has resulted in 30 trials per fairness level (0, 25, 50, 75, 100), 15 paired with expected and 15 with unexpected melodies.

- *Are all trials the same length in duration? To avoid participants to finish the task quicker by choosing "leave".*

We have now specified that a screen asking to "Wait for the next trial" appears in case participants decide to Leave. This screen lasts three seconds, which is how long it takes, on average, to select the amount to spend.

- *Some suggestions for the instructions in Appendix A:*

o The instructions of the role of the players in the game are not very clearly introduced before explaining the actions the player can take. Specifically, line 25 where it explains the actions of player A without any context of the role of player A and the game. I would suggest to include some sort of cover story as to why player A can steal from B. For example, by introducing that player A has given the possibility or power to take some of the divided chips from player B to increase player A's total amount of chips (and money). At the same time, player B cannot do anything. Then introduce player C's role.

We have now added a more explicit short cover story in the instructions (Appendix), with some examples in order to clarify the rules.

o The introduction of the music part comes a bit unexpected and is quite obvious that it probably should have an effect on their behaviour. Perhaps the authors could try to better fit this in the game with a cover story so it doesn't become so obvious.

This is a very good point and we thank the reviewer for raising it. We have now added a short explanation as to why melodies will be played during the task in the Participants Information Sheet, which is the first document that participants read once in the lab. We are going to verbally re-iterate the presence of music and its scope in the experiment before starting the task.

"The study aims to explore whether listening to music interferes with the brain processes involved in making decisions. As you listen to some music through earplugs, you will be presented with a variety of scenarios and will be required to make decisions on how to spend your allocated money in each scenario"

o The Take action becomes a bit confusing with the "How much do you want to spend" screen, as in this part it is not clear how much 30 chips spend means. I would suggest to better integrate the take action, spending of own chips and how that affects

punishment (x3) in the text before showing an example of the trial.

We have now added a more detailed explanation of the Take action in the instruction. Moreover, we have added more information before the practice trials, repeating the main rules of the game.

References

Pearce, M.T. (2005). The construction and evaluation of statistical models of melodic structure in music perception and composition. PhD thesis, London, UK: Department of Computing, City University.

Pearce, M. T., & Wiggins, G. A. (2012). Auditory expectation: the information dynamics of music perception and cognition. *Topics in cognitive science*, 4(4), 625-652.

Pearce, M. T., Herrojo Ruiz, M., Kapasi, S., Wiggins, G. A., & Bhattacharya, J. (2010). Unsupervised statistical learning underpins computational, behavioural and neural manifestations of musical expectation. *NeuroImage*, 50(1), 303–314.

Appendix C

Editor Comments to Author (Professor Chris Chambers):

The two original reviewers responded positively to the revision. There is one remaining statistical issue to consider from Reviewer 2. Once this is addressed in a response, Stage 1 acceptance should be forthcoming without requiring further in-depth review.

We thank the reviewer and the Editor for the comments provided, and we look forward to start data collection!
Please see our responses below.

Reviewer comments to Author:

Reviewer: 1

Comments to the Author(s)

Dear Elisa and Claudia,

my compliments for the revision. I still think the music violations are not violating much. They actually sound more interesting than the rest :-)
Perhaps in naive listeners they may sound more violating.

Good luck with the data collection.

Thank you, and thanks for your useful comments.

Reviewer: 2

Comments to the Author(s)

I believe the authors have addressed all my concerns and comments in an adequate way.

I have one minor comment on the mixed effect model in the r-script:

Once the authors have collected the data, I would suggest that when running mixed effects models, they first test whether their mixed effect models should at least include by-participant random slopes for the within-subject predictors (Fairness expectation, Music expectancy). They can do this by performing model comparison (with `anova()` function), to justify what their maximal model is given their data. At the moment in their r-script, the models are all random-intercept only models. The authors can do the model comparison for all combinations of random slopes. The same holds for the random intercept of Block, if the authors expect random variation in slopes for Division and InfCont by block.

Thank you for suggesting this step in the data analysis. We have now included random slopes for the two predictors, followed by a model comparison to identify the best maximal

model. We do not expect random variation in slopes for Division and InfCont by block, so we have not included a model that accounts for this.

The information has been added to the r script and an example of this has been copied below for you to see for the *Choice DV*.

```
Choice (binary) - random intercept
glmer_cho_1 <- glmer(Choice ~ Division * InfCont + Fair_exp + (1 | Subject) + (1 | Block),
family="binomial"(link = "logit"), data=EEGExp_behav_panel,
control=glmerControl(optimizer="bobyqa",optCtrl=list(maxfun=100000)))
options(scipen=999)
summary (glmer_cho_1)
```

```
Extended model choice 2 - slopes
glmer_cho_2 <- glmer(Choice ~ Division * InfCont + Fair_exp + (1 | Subject) + (1 | Block)
+ (1 + Division | Subject) + (1 + InfCont | Subject), family="binomial"(link = "logit"),
data=EEGExp_behav_panel,
control=glmerControl(optimizer="bobyqa",optCtrl=list(maxfun=100000)))
options(scipen=999)
summary (glmer_cho_2)
```

```
Extended model choice 3 - slope Division
glmer_cho_3 <- glmer(Choice ~ Division * InfCont + Fair_exp + (1 | Subject) + (1 | Block)
+ (1 + Division | Subject), family="binomial"(link = "logit"), data=EEGExp_behav_panel,
control=glmerControl(optimizer="bobyqa",optCtrl=list(maxfun=100000)))
options(scipen=999)
summary (glmer_cho_3)
```

```
Extended model choice 4 - slope InfoContent
glmer_cho_4 <- glmer(Choice ~ Division * InfCont + Fair_exp + (1 | Subject) + (1 | Block)
+ (1 + InfCont | Subject), family="binomial"(link = "logit"), data=EEGExp_behav_panel,
control=glmerControl(optimizer="bobyqa",optCtrl=list(maxfun=100000)))
options(scipen=999)
summary (glmer_cho_4)
```

```
### ANOVA ON THE MODELS
anova(glmer_cho_1, glmer_cho_2)
```

```
anova(glmer_cho_1, glmer_cho_3)
```

```
anova(glmer_cho_1, glmer_cho_4)
```

```
anova(glmer_cho_2, glmer_cho_3)
```

```
anova(glmer_cho_2, glmer_cho_4)
```

References:

1. Matuschek H, Kliegl R, Vasishth S, Baayen H, Bates D (2017) Balancing Type I error and power in linear mixed models. *J Mem Lang*. doi:10.1016/j.jml.2017.01.001.
2. Bates D, Kliegl R, Vasishth S, and Baayen, H. (2015). "Parsimonious Mixed Models." ArXiv:1506.04967. <http://arxiv.org/abs/1506.04967>.
3. Barr DJ, Levy R, Scheepers C, Tily HJ (2013) Random effects structure for confirmatory hypothesis testing: Keep it maximal. *J Mem Lang* 68(3):255–278.

I would like to wish the authors good luck with their study.

Thank you!

Appendix D

Dear Reviewers,

We have finally completed the Stage 2 process of the registered report titled “Does unfairness sound wrong? A cross-domain investigation of expectations in music and social decision-making”.

The collected data and code used for the analyses have been uploaded on Dryad, with information about the data. Material related to the approved Stage 1 protocol are available on the Open Science Framework. The links for both can be found on page 7 of the manuscript. Please note, however, that we have encountered an issue with the upload process on Dryad. For this reason, to speed up the process, we have also added a temporary Dropbox link for you, which you can find at the bottom of this page. Once we fix the issue, we will add a permanent public link from Dryad where all data and code will be available.

We confirm that the experiment was executed and analysed in the manner originally approved. To aid the reviewing process, minor changes have been noted and highlighted in yellow in the manuscript. These include:

- the future tense in the Methods section was changed to the past tense;
- a minor inconsistency regarding the number of trials was spotted in the Methods section that had been submitted for Stage 1; this has now been corrected;
- the MARA pre-processing stage was fully automated for replicability and objectivity, and this has now been noted in the manuscript and commented in the Matlab code;
- minor stylistic changes to the code were made to improve the code’s efficiency, and these have been noted in the relevant Matlab code. However, these changes did not affect the analysis protocol agreed at Stage 1.

Following the pre-registered analyses, we have also included some follow-up exploratory analyses, which you can find in a separate section in the Results.

Once again, we thank you for the important contributions and suggestions you made to the study and manuscript. We hope you enjoy reading the revised manuscript for Stage 2 and we look forward to receiving your comments in due time.

Sincerely,

Dr Elisa Carrus and Dr Claudia Civai

Data access and usage notes

Download the folder Dryad from:

<https://www.dropbox.com/sh/5c4mprke4cec4bb/AABcWasFgss-GZIJ34XCXdgua?dl=0>

The EEG data folder contains:

- Raw unprocessed data in EEGLab format (.set)

- Processed data at the final stage of pre-processing (after MARA step - see Matlab code)
- Group data in Fieldtrip format for plotting and statistics

The behavioural data contains:

- Raw data in .csv format
- Processed data for all subjects in .csv format
- Group data in .csv format for factorial analysis and plots in

The code folder contains the .m MATLAB file for EEG analysis. This contains comments of what is done at each step, plus minor changes compared to Stage 1 submission. The R code for the model analyses (read raw data and creates group data).

Note on conditions (these can also be found in the MATLAB code)

Factor 1: melodic expectancy (Expected note: 1, Unexpected note: 2)

Factor 2: fairness division (fair (0)= 1, 25 = 2, 50 =3, 75 =4, 100 =5).

Appendix E

Dear Associate Editor and Reviewer,

We are very pleased with the decision and we thank you for returning your comments so quickly.

We have now made all the changes requested, and you can find our responses below.

We have very much enjoyed working on the Registered Report and have certainly seen the benefits of it first-hand. We also want to take this opportunity to thank the Journal for offering this format of submission and for the support offered at all stages of this process.

Best wishes

Claudia, Rachel and Elisa

Author responses to comments

Associate Editor Comments to Author (Professor Chris Chambers):

Associate Editor: 1

Comments to the Author:

One of the previous Stage 1 reviewers was available to assess the Stage 2 manuscript and I have also read it myself. I agree with the reviewer's positive assessment and also the critical points -- primarily that the conclusions must be dominantly shaped by the outcomes of the preregistered hypothesis tests, with the distinction between preplanned and exploratory outcomes made clear at all times (particularly in the Abstract). In all other respects the submission is a fine exemplar of the RR format. Following minor but careful revision to address this point throughout the manuscript, full acceptance is likely to be forthcoming without requiring further in-depth review.

Thank you, we agree with this. We have made sure that the distinction between exploratory and planned analyses is clear throughout, and the discussion and conclusions have been rewritten to be predominantly driven by the planned analyses.

Comments to Author:

Reviewer: 1

Comments to the Author(s)

I read the paper by Claudia Civali et al. and I think authors did a great job. Congratulations. It is not easy to collect so many data with a EEG study!

Thank you very much!

That said. I have still a few comments. Overall, it looks to me authors are still very much in love with the original hypothesis. Although results do not show the hypothesized effect, authors discuss "marginally significant" results and make speculations about possible trends. These speculations, in my opinion, should be dropped all together. Here we have a solid study that had

the statistical power (ie the fuel) to detect effects of certain sizes (ie to travel certain distances). There is simply not enough fuel to travel the various possible effects the authors see in the results.

Speculations have been removed from the discussion and the relevant sections have been highlighted in the new manuscript submitted.

One additional point. I'm not an "effect size" expert, but If you look at the formula of the partial eta square, it ranges from 0 to 1. Effects sizes in the range of 0.05 looks to me very small...

We used Cohen's benchmarks to define the size of the effect size as explained in Lakens, D. (2013). "Cohen (1988) has provided benchmarks to define small ($\eta^2 = 0.01$), medium ($\eta^2 = 0.06$), and large ($\eta^2 = 0.14$) effects." (Lakens, 2013, p. 7). See also <https://imaging.mrc-cbu.cam.ac.uk/statswiki/FAQ/effectSize>

A few specific comments:

- Abstract. Please write the abstract in such a way that is crystal clear to the reader the distinction between hypothesis driven analysis/results and exploratory results. There should be no confusions when reading the abstract.

We have modified the abstract accordingly

Results section. Drop "marginally significant" for $p = .06/.07$ etc. There is not such a thing like a marginally significant result. Drop also words like "trend" or "tendency".

We have changed 'marginally significant' to 'not significant' for any p-values equal or higher than p.06. We changed 'marginally significant' to 'significant' for the effect of melodic expectancy on Choice ($p = .053$) and changed to $p = .05$. We changed 'marginally significant' to 'not significant' for the effect of the covariate of fairness expectation on Choice ($p = .056$) and changed to $p = .06$. We removed any reference to trend or tendency. Relevant sections have been highlighted in yellow.

Figures. Please convert barplots into boxplots with mean on top

We have converted these as required.

References

Lakens, D. (2013). Calculating and reporting effect sizes to facilitate cumulative science: a practical primer for t-tests and ANOVAs. *Frontiers in psychology*, 4, 863.